# Wayward Concepts In Multimodal Models

**Brandon Trabucco**[1]**, Max Gurinas**[2]**, Kyle Doherty**[3]**, Ruslan Salakhutdinov**[1]
[1]Carnegie Mellon University, [2]University Of Chicago Laboratory Schools, [3]MPG Ranch
`brandon@btrabucco.com, rsalakhu@cs.cmu.edu`

## ABSTRACT

Large multimodal models such as Stable Diffusion can generate, detect, and classify new visual concepts after optimizing just the prompt. How are prompt embeddings for visual concepts found by prompt tuning methods different from typical discrete prompts? We conduct a large-scale analysis on three state-of-the-art models in text-to-image generation, open-set object detection, and zero-shot classification, and find that prompts optimized to represent new visual concepts are akin to an adversarial attack on the text encoder. Across 4,800 new embeddings trained for 40 diverse visual concepts on four standard datasets, we find perturbations within an $\epsilon$-ball to any prompt that reprogram models to generate, detect, and classify arbitrary subjects. These perturbations target the final-layers in text encoders, and steer pooling tokens towards the subject. We explore the transferability of these prompts, and find that perturbations reprogramming multimodal models are initialization-specific, and model-specific. Code for reproducing our work is available at the following site: wayward-concepts.github.io.

## 1 INTRODUCTION

Fine-tuning prompts is a widely successful technique for adapting large pretrained models to new tasks from limited data (Lester et al., 2021; Li & Liang, 2021; Shin et al., 2020; Gal et al., 2023). In language modelling, these prompts can efficiently teach pretrained language models specialized tasks, such as reading tables (Li & Liang, 2021). In text-to-image generation, they can embed subjects with unique, often hard-to-describe appearances into the generations of a diffusion model (Gal et al., 2023; Ruiz et al., 2023). Large multimodal models, such as Stable Diffusion (Rombach et al., 2022), OWL-v2 (Minderer et al., 2022), and CLIP (Radford et al., 2021; Cherti et al., 2023; Fang et al., 2023), can generate, detect, and classify diverse visual concepts not present in their training data after fine-tuning embeddings representing that concept in their prompt (Gal et al., 2023; Trabucco et al., 2023). *How do soft prompts obtained via prompt tuning methods that encode specific visual concepts (i.e. black dog) differ from typical discrete prompts?* There is a popular hypothesis in multimodal machine learning that text-based models have emergent representations for visual content (Liu et al., 2023a; Koh et al., 2023; Huh et al., 2024; Lu et al., 2022), despite training purely on text. This investigation aims to determine if visual prompt tuning methods find emergent embeddings akin to the existing discrete prompts, or if they find something more subtle.

We conduct a large-scale study on three state-of-the-art models in text-to-image generation, open-set object detection, and zero-shot classification. We optimize 4,800 new embeddings for Stable Diffusion (Rombach et al., 2022), OWL-v2 (Minderer et al., 2022), and CLIP (Radford et al., 2021; Cherti et al., 2023; Fang et al., 2023) to generate, detect, and classify 40 diverse visual concepts across four standard datasets. For all tested models and datasets, visual prompt tuning finds solutions within an $\epsilon$-ball to unrelated discrete prompts that reprogram models to generate, detect, and classify arbitrary subjects. We refer to this redundancy of solutions as **fracturing** of the embedding space. Fractured solutions found by visual prompt tuning have noteworthy properties: first, solutions are robust to their location in the embedding space, and second, they target specific layers in the model.

Across all models and tasks, fractured solutions anchored to unrelated discrete prompts perform comparably to solutions anchored to related discrete prompts. In both cases, fractured solutions target the final layers of text encoders to steer text encoder representations away from the original discrete prompt, and towards the desired subject. We explore the transferability of solutions found by visual prompt tuning, and find the perturbations are specialized to the model, and anchor prompt.

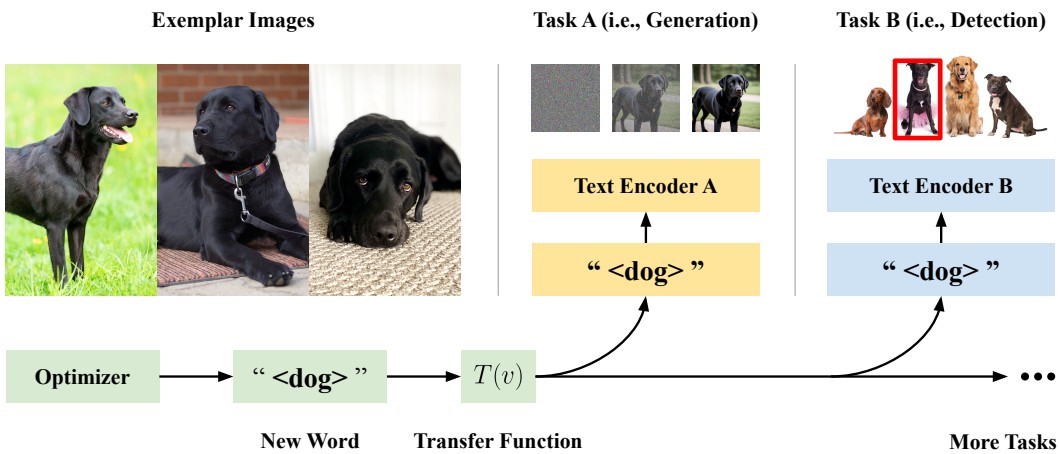

Figure 1: Large multimodal models can learn specific concepts, such as black Labrador retriever in the figure, after fine-tuning just the prompt. How do soft prompts obtained via prompt tuning methods that encode specific visual concepts (i.e. black dog) differ from typical discrete prompts? We study the solutions found by prompt tuning methods on three models, and show their solutions are akin to adversarial attacks on text encoders.

Indeed, the perturbations found by visual prompt tuning are akin to adversarial attacks (Goodfellow et al., 2015; Brown et al., 2017), on the embedding space of the text encoder.

This work contributes a large-scale study of soft prompts that encode specific visual concepts across generation, detection, and classification tasks. We provide a benchmark for training such prompts on a diverse set of visual concepts, and evaluating their transferability across three models. Our work aims to galvanize the interoperability of large multimodal models following Figure 1, allowing prompts trained for generating black Labradors to be re-used for detection, and other tasks. Transferring prompts can significantly improve the adaptability and cost of machine learning systems by eliminating the need to re-train prompts when new models are released. We highlight the difficulty of transferring prompts for current models, investigate what these prompts learn, and what prevents them from transferring successfully. Code for reproducing our work, and benchmarking new transfer methods is available at the following official website: wayward-concepts.github.io.

## 2 RELATED WORKS

**Text-To-Image Generation.** With the advent of diffusion-based architectures, large-scale generative models have developed impressive photo-realism. Approaches like Stable Diffusion (Rombach et al., 2022), DALL-E 2 (Ramesh et al., 2022), and Imagen (Saharia et al., 2022) employ diffusion-based approaches (Ho et al., 2020; Sohl-Dickstein et al., 2015) that start from an initial Gaussian noise map, and iteratively denoise the image over several denoising diffusion steps. These approaches incorporate pretrained text-encoders, such as CLIP (Radford et al., 2021) in Stable Diffusion (Rombach et al., 2022), to guide generation in the diffusion process. Guidance is typically applied through Classifier-free Guidance (Ho & Salimans, 2022), which allows the influence of the text-encoder to be increased, at the expense of generation quality. Diffusion models have remarkable flexibility, and can generate new subjects from a handful of examples by learning embeddings for pseudo tokens representing the subject in the prompt (Gal et al., 2023; Trabucco et al., 2023). Fine-tuning both the model and the prompt, as in Dreambooth (Ruiz et al., 2023), leads to improved generation of subjects, while retaining the controllability of pseudo tokens. Pseudo tokens in diffusion models are an application of visual prompt tuning, and our investigation considers them.

**Open-Vocabulary Object Detection.** Parallel to work in generation, large-scale object detection models have developed a comparable strong versatility, and can detect new objects from short descriptions of their appearance (i.e. detect *black dog*) (Zou et al., 2023b; Liu et al., 2023b; Minderer et al., 2022). Models like Grounding DINO (Liu et al., 2023b), SEEM (Zou et al., 2023b), and

OWLv2 (Minderer et al., 2022) employ a pretrained text encoder to produce representations for classifying bounding boxes. In OWLv2 (Minderer et al., 2022), representations from a pretrained CLIP (Radford et al., 2021) text encoder are contrasted with region-based representations from a vision transformer backbone. Grounding DINO (Liu et al., 2023b), and SEEM (Zou et al., 2023b) employ representations from a pretrained text encoder (BERT (Devlin et al., 2019), and UniCL (Yang et al., 2022), respectively) to directly guide bounding box proposal. We show open-vocabulary object detectors can detect new objects from a handful of examples by optimizing just new prompt embeddings for the object. Our analysis shows key properties of these optimized prompts in open-vocabulary object detection are shared with text-to-image generation.

**Zero-Shot Classification.** We use recent models (Fang et al., 2023) derived from CLIP (Radford et al., 2021) for zero-shot classification experiments. We insert new word embeddings optimized for classifying new visual concepts in the prompt of the CLIP text encoder, and contrast text representations with image representations from the CLIP vision encoder on test images. Prior work shows CLIP is an effective zero-shot classifier (Radford et al., 2021; Novack et al., 2023). We use checkpoints from OpenAI CLIP (Radford et al., 2021), OpenCLIP (Cherti et al., 2023), and Data Filtering Networks (Fang et al., 2023), trained on LAION-5B (Schuhmann et al., 2022). Diffusion models can also be used as zero-shot classifiers (Li et al., 2023; Clark & Jaini, 2023), but we focus on CLIP for better task coverage. Our analysis shows that CLIP has a fractured embedding space, and visual prompt tuning for zero-shot classification finds fractured solutions.

**Prompt-Tuning.** The word embeddings we optimize for visual concept learning tasks are closely related to prompt-tuning (Lester et al., 2021; Li & Liang, 2021). Prompt tuning aims to find a prefix or an entire prompt that causes a pretrained language model to perform a specialized task, such as reading tables (Lester et al., 2021; Li & Liang, 2021). These methods treat the prompt as a trainable parameter, and optimize the embeddings of the prompt to minimize a task loss function. Prior work has shown the resulting soft prompts in language modelling tasks are hard to interpret (Khashabi et al., 2022), as their closest discrete prompts are often unrelated to the desired task. Transferring learned prompts is an important task in jail-breaking LLMs (Zou et al., 2023a; Robey et al., 2023), and researchers are searching over discrete prompts (Wen et al., 2023; Shin et al., 2020; Zou et al., 2023a; Robey et al., 2023). In pure language modelling tasks, researchers have shown that certain soft prompts can transfer between models with the same architecture and task, but different weights (Passigan et al., 2023; Ju et al., 2023; Wu et al., 2023). We extend this investigation to visual tasks, models with different label modalities (images, bounding boxes, and class labels).

**Adversarial Examples.** Perturbations found by visual prompt tuning near discrete prompts are akin to adversarial attacks on the embeddings of the text encoder. Adversarial robustness is an extensively studied field in computer vision (Goodfellow et al., 2015), with a variety of attack methods, including (Goodfellow et al., 2015; Brown et al., 2017; Madry et al., 2018; Kurakin et al., 2017; Carlini & Wagner, 2017), and defense methods, including (Madry et al., 2018; Qin et al., 2019; Tramèr et al., 2018; Kannan et al., 2018). Adversarial attacks in computer vision traditionally focus on modifying the pixels in an image, whereas we modify word embeddings. Adversarial attacks on language include jail-breaking (Zou et al., 2023a; Robey et al., 2023), and typically involve searching over discrete prompts (Zhang et al., 2020; Li et al., 2020), rather than continuous embeddings.

## 3 TRANSFER EVALUATION METHODOLOGY

Transferring prompt tuning solutions involves finding a map between the embedding spaces of different models. We call this mapping the Transfer Function $T(v)$, depicted in Figure 2. The Transfer Function maps word representations for visual concepts from the vector space $\mathcal{X} = \mathbb{R}^{d_x}$ for word embeddings in one model, to the vector space $\mathcal{Y} = \mathbb{R}^{d_y}$ for word embeddings in another model. Space $\mathcal{X}$ may correspond to Stable Diffusion Rombach et al. (2022) word embeddings for a generation task, and $\mathcal{Y}$ may be OWL-v2 Minderer et al. (2022) word embeddings for a detection task. Given these vector spaces, the Transfer Function predicts the representation $\vec{x}(w)$ in the vector space $\mathcal{X}$ for a word $w$ originally from the vector space $\mathcal{Y}$ given just the word vector representation $\vec{y}(w)$.

$$T^{\,y\to x} : \mathcal{Y} \to \mathcal{X} = \arg\min_T \ \mathbb{E}_{w\sim p_w} \|\vec{x}(w) - T(\vec{y}(w))\|_2^2 \tag{1}$$

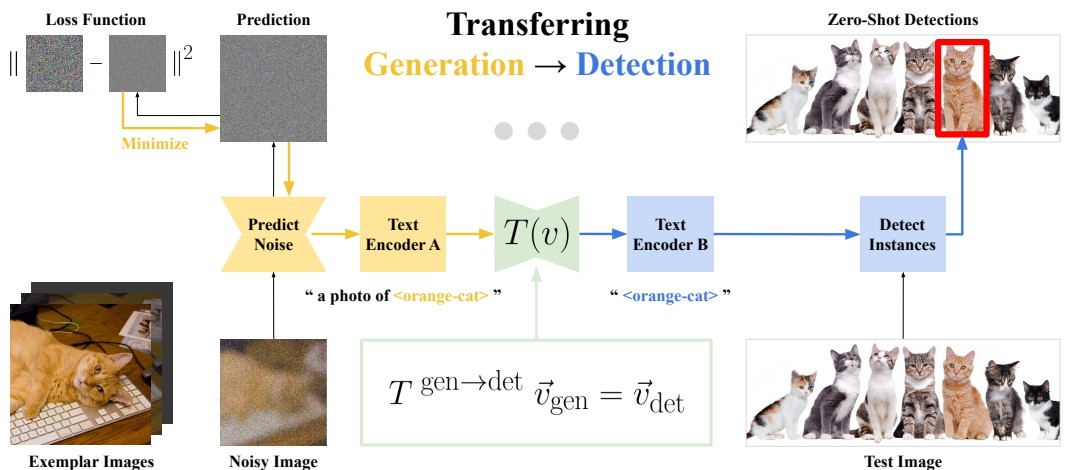

Figure 2: Transferring prompt embeddings from generation to detection tasks. We fine-tune the vector embeddings for new tokens (such as `<orange-cat>` for the orange cat in the figure) to minimize a noise prediction loss for generation. Vector embeddings are transferred from generation to detection using a linear map, and used to produce zero-shot instance detections for the target visual concept (in this case, orange cats).

The Transfer Function $T(v)$ minimizes the average prediction error between transferred word embeddings $T(\vec{y}(w))$ and real word embeddings $\vec{x}(w)$ from the vector space $\mathcal{X}$. We average this error over a uniform distribution $p_w$ of words that exist in both vector spaces $\mathcal{X}$ and $\mathcal{Y}$. In our experiments on Stable Diffusion 2.1 Rombach et al. (2022) and OWL-v2 Minderer et al. (2022), the number of words in $p_w$ ($> 40,000$) is larger than the number of components $d_x$ and $d_y$ in each vector space.

## 3.1 FINDING A LINEAR TRANSFER FUNCTION

Solving the optimization problem given by Equation 1 is hard in general, and to simplify the investigation, we restrict our focus to the class of linear Transfer Functions. This restriction transforms the hard problem in Equation 1 into a Linear Least Squares estimator, which has a closed-form solution. Consider a pair of matrices $X \in \mathbb{R}^{n \times d_x}$ and $Y \in \mathbb{R}^{n \times d_y}$, where each pair of rows in $X$ and $Y$ is a pair of word vector embeddings $\vec{x}(w)$ and $\vec{y}(w)$ for a word $w$ contained in the support of the distribution $p_w$. The Linear Least Squares estimator we employ for $T^{\,y \rightarrow x}$ is given below.

$$T^{\,y \rightarrow x} = \arg\min_{T} \; \mathbb{E}_{w \sim p_w} \|\vec{x}(w) - T\vec{y}(w)\|_2^2 = (Y^T Y)^{-1} Y^T X \tag{2}$$

One can interpret $T^{\,y \rightarrow x}$ as lining up the directions in the vector spaces $\mathcal{X}$ and $\mathcal{Y}$ that correspond to the same visual concepts. Word embeddings often have algebraic relationships (Mikolov et al., 2013), and a linear Transfer Function preserves these relationships by distributing over addition.

## 3.2 EVALUATING PROMPTS ON TRANSFERRED TASKS

Using Equation 2, we estimate Transfer Functions between all six ordered subsets of the three models, and evaluate prompts optimized for visual concepts on one task (such as generation), and transferred to the same visual concepts on another task (such as classification). Consider a dataset $D$ of images $I$ depicting a specific visual concept, such as a black Labrador retriever, and task-specific annotations $a_y$, such as bounding boxes ($a_y \in \mathbb{R}^{b \times 4}$), or class labels ($a_y \in \mathbb{N}$). We first optimize the prompt embeddings $\vec{v}_y \in \mathcal{Y}$ to minimize a task-specific loss function $\mathcal{L}_y$. We then zero-shot transfer embeddings $\vec{v}_y$ to task $x$ using the linear map $\vec{v}_x = T^{\,y \rightarrow x} \vec{v}_y$, and evaluate a task-specific performance metric $\mathcal{M}_x$. Loss functions and performance metrics for each task are shown in Table 1.

$$\mathbb{E}_{I, a_x \sim D_{\text{test}}} \; \mathcal{M}_x(T^{\,y \rightarrow x} \vec{v}_y, I, a_x) \;\; \text{s.t.} \;\; \vec{v}_y = \arg\min_{\vec{v}} \; \mathbb{E}_{I, a_y \sim D_{\text{train}}} \; \mathcal{L}_y(\vec{v}, I, a_y) \tag{3}$$

For Stable Diffusion 2.1 (Rombach et al., 2022), we use the denoising loss function originally proposed in Ho et al. (2020), where the goal is to predict a noise map $\epsilon$ added to an image $I$ at a particular

| Task | Loss Function | Performance Metric |
|---|---|---|
| Generation | $\mathbb{E}_{I \sim D_{\text{train}}} \\|\epsilon - \epsilon_\theta(\sqrt{\alpha_t}I + \sqrt{1 - \alpha_t}\epsilon, t, \vec{v})\\|^2$ | $\mathbb{E}_{I \sim p_\theta(\cdot \mid \vec{v})} \mathbb{1}[\, I \text{ has the concept }]$ |
| Detection | $\mathbb{E}_{I,b,w \sim D_{\text{train}}} [w \cdot (\vec{e}_{\text{object}}(I, b)^T \vec{e}_{\text{text}}(\vec{v}))]$ | Mean Average Precision |
| Classification | $\mathbb{E}_{I,w \sim D_{\text{train}}} [w \cdot (\vec{e}_{\text{image}}(I)^T \vec{e}_{\text{text}}(\vec{v}))]$ | Classifier Accuracy |

Table 1: Loss Functions and Performance Metrics. We benchmark transfer of prompt embeddings across generation, detection, and classification tasks. In each row, $I$ corresponds to an image, $b$ to an object bounding box, and $w \in \{-1, 1\}$ to a weight multiplied onto the loss function. This weight controls whether the objective is maximized or minimized, where $w = -1$ when the image and bounding box contain the target concept, and $w = 1$ otherwise. The functions $\vec{e}$ are image and text encoders that return vector representations: $\vec{e}_{\text{image}}$ is the CLIP vision encoder, $\vec{e}_{\text{text}}$ is the CLIP text encoder, and $\vec{e}_{\text{object}}$ is an OWL-v2 region proposal feature.

timestep in the diffusion process $t$. We optimize the prompt embeddings $\vec{v}$ so that Stable Diffusion generates images of a particular class (such as black Labrador). This optimization uses a training dataset $D_{\text{train}}$, and a separate dataset $D_{\text{test}}$ that contains different images of the same visual concept (such as black Labrador) is used for evaluation. For evaluating generation, we measure the probability that generations contain the target visual concept, measured by OpenAI's pretrained CLIP L-14 model given the prompt "a photo of {subject_name}". This procedure employs Textual Inversion (Gal et al., 2023) with the additional prompt embedding transfer step in Figure 2.

We use standard loss functions and performance metrics adapted from recent literature when training and evaluating visual prompts. Each loss function and performance metric is discussed further in Section 4.3. Now equipped for training, evaluating, and transferring prompts, we can pose our motivating question: *how do solutions from prompt tuning differ from typical discrete prompts?*

## 4 Prompt Tuning Finds Fractured Solutions

Near any discrete prompt in the embedding space of Large Multimodal Models, there are perturbations that reprogram models to generate, detect, and classify arbitrary subjects. For example, the prompt tuning solution in the top-left of Figure 3 is closest to the discrete prompt "cat", but Stable Diffusion generates a vase. We observe this high degree of model reprogrammability consistently across three model classes, four standard datasets, and 40 diverse visual concepts, suggesting a general phenomenon in Large Multimodal Models. We refer to this phenomenon as **fracturing** of the embedding space, as prompts that encode (i.e. generate) specific visual concepts are scattered across the entire embedding space. Figure 3 shows example solutions—each row corresponds to a visual concept from a standard dataset, and each column is a discrete anchor prompt. In several cases, an *identical image* is generated by several perturbations near different anchors, such as generations for the duck concept (second row) from the vase (column two) and candle anchors (column four).

### 4.1 Dataset Preparation

We explore the fracturing phenomenon using four standard datasets, adapted from recent literature in generation, detection, and classification. We adapt the 2014 ImageNet detection dataset (Deng et al., 2009), the DreamBooth dataset (Ruiz et al., 2023), COCO (Lin et al., 2014), and PASCAL VOC (Everingham et al., 2010). For each dataset, we select 10 concepts uniformly at random from available classes to use for benchmarking, and select 8 images per concept from the training set (see Appendix D). These cover a diverse range of concepts likely to be encountered in real use cases. For ImageNet, each image is annotated with an integer class label, and a set of bounding boxes that contain the target concept. For the DreamBooth dataset, bounding box labels are missing. To obtain bounding box labels, we ran a pretrained OWL-v2 on every image using the name of the subject as the prompt, and manually verified the labels as correct. For COCO and PASCAL VOC, class labels are not present, so we assign each image a class label equal to the class of the largest bounding box.

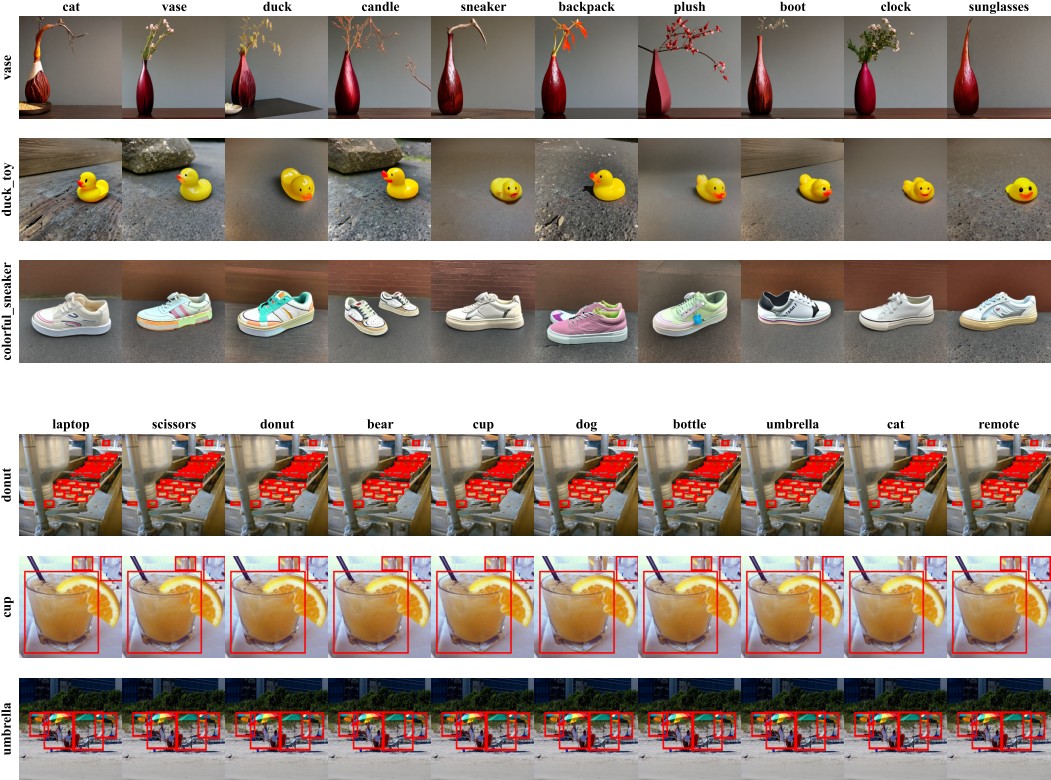

Figure 3: Example generations and detections for various concepts (row labels) using solutions found in the immediate neighborhood of unrelated words (column labels). We consistently find soft prompts for generating and detecting arbitrary concepts near unrelated discrete prompt anchors across ImageNet (examples in Appendix J), DreamBooth (first three rows), COCO (last two rows), and PASCAL VOC (Appendix J) datasets. The same objects are detected, and in several cases, near-identical images are generated for different anchors.

## 4.2 MODEL DETAILS

We select three state-of-the-art models in text-to-image generation, open-set object detection, and zero-shot classification. Each model accepts a text-based prompt as input, containing the prompt embeddings to be optimized (i.e. "<dog>" for the dog concept). For generation, we select Stable Diffusion 2.1, a latent diffusion model proposed by Rombach et al. (2022). For detection, we select OWL-v2 (Minderer et al., 2022), a two-stage object detection model with a region proposal stage, and a classification stage that labels region proposals with classes. For classification, we select Data Filtering Networks (Fang et al., 2023), which use CLIP-based language-image contrastive learning (Radford et al., 2021) on a filtered dataset. These models have different image size requirements. We resize images to 768x768 pixels for Stable Diffusion (Rombach et al., 2022), 960x960 pixels for OWL-v2 (Minderer et al., 2022), and 224x224 for Data Filtering Networks (Fang et al., 2023).

## 4.3 EXPERIMENT DETAILS

**Training** We take all combinations of models, datasets, and concepts, and perform 10 randomized trials varying the discrete prompt anchor to initialize prompt tuning. Anchors are selected as the closest single token in the model's tokenizer to the name of the target anchor concept. For example, "sombrero" tokenizes to multiple subwords, so we use 'hat' as the anchor. This choice ensures the experiments account for a broad range of initializations. We optimize the prompt embeddings for each concept using the Adam (Kingma & Ba, 2015) optimizer with a learning rate of 0.0001, and a batch size of 8 (these hyperparameters are shared across all models). We train for 1000 gradient descent steps, and report performance metrics using the final optimized prompt embeddings.

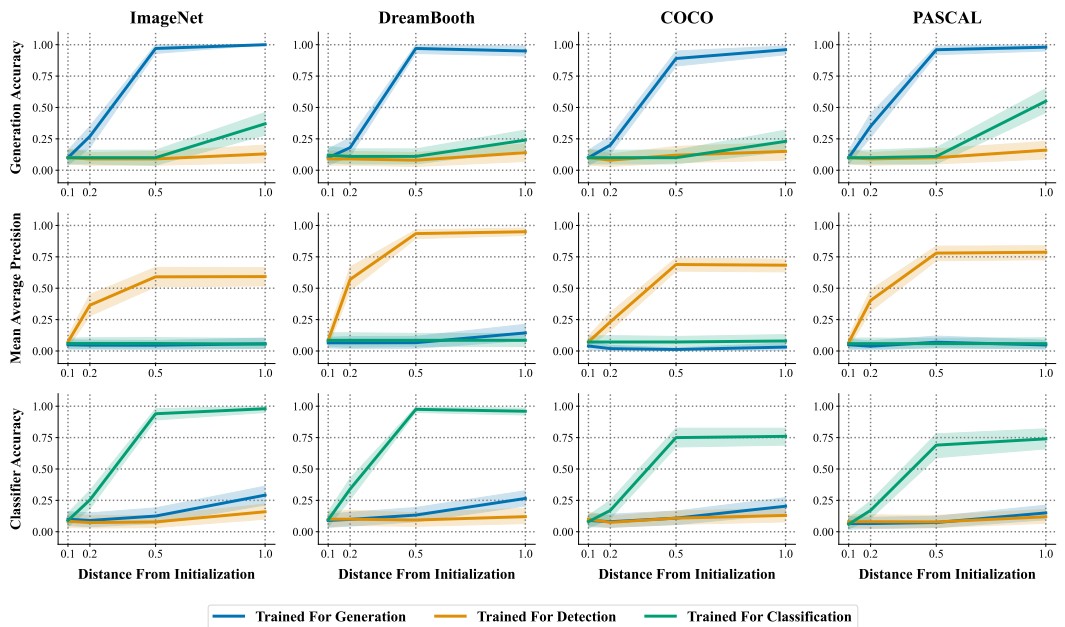

Figure 4: Performance (y-axis) of soft prompts optimized to cause generation, detection, and classification of new visual concepts, for different constraint levels (x-axis). In-domain performance saturates at a constraint level of $\delta = 0.5$, which corresponds to solutions where the nearest existing word vector is the anchor $w_{\text{anchor}}$. Constrained solutions perform well in-domain, but typically don't perform well on transferred tasks for $\delta < 1$. Each line in the figure corresponds to the 95% confidence interval of 100 randomized trials for 10 concepts, and 10 anchor words per dataset. Refer to Appendix D for the concepts and anchor words used for each dataset.

**Loss Functions** For generation, we employ the standard reparameterized denoising objective, introduced by Ho et al. in DDPM (Ho et al., 2020). For detection, we maximize the cosine similarity between the text and region feature containing the target object, and minimize cosine similarity to all other region features proposed by OWL-v2 (Minderer et al., 2022) in the image. For classification, we maximize cosine similarity between text and images of the target concept, and minimize cosine similarity to images that don't contain the target concept. Table 1 shows the exact loss definitions.

**Metrics** For evaluating the quality of generations, we report the rate at which a pretrained OpenAI CLIP L-14 (Radford et al., 2021) classifier predicts that generations are the target class, labeled Generation Accuracy in Table 1. The label set for Generation Accuracy is the set of all concepts from the corresponding dataset (names listed in Appendix D). For detection, we report the Mean Average Precision of OWL-v2 (Minderer et al., 2022) bounding box predictions on a held-out validation set. For classification, we report the accuracy of DFN CLIP (Fang et al., 2023; Radford et al., 2021) given images of all concepts (both positive and negative examples) from a held-out validation set. All metrics are reported as 95% confidence intervals over 10 randomized trials varying the anchor.

## 4.4 CONTROLLING THE LOCATION OF SOLUTIONS

Large Multimodal Models are highly reprogrammable. We explore this phenomenon by considering a constrained objective for soft prompts in Equation 4, where given an anchor token $w_{\text{anchor}}$ used to initialize $\vec{v}$, and a threshold $\delta$ normalized by the distance of the closest discrete token to the anchor, we constrain solutions for $\vec{v}$ to an l2-ball of radius $\delta$, implemented using projected gradient descent. We conduct a large-scale experiment, optimizing 4,800 prompt embeddings for 40 visual concepts across four standard datasets, three models, and four constraint thresholds $\delta \in \{0.1, 0.2, 0.5, 1.0\}$.

$$\vec{v}_y = \arg\min_{\vec{v}} \ \mathbb{E}_{I, a_y \sim D_{\text{train}}} \ \mathcal{L}_y(\vec{v}, I, a_y) \ \text{ s.t. } \ \frac{\|\vec{v} - y(w_{\text{anchor}})\|_2}{\min_{w \neq w_{\text{anchor}}} \|y(w) - y(w_{\text{anchor}})\|_2} \leq \delta \qquad (4)$$

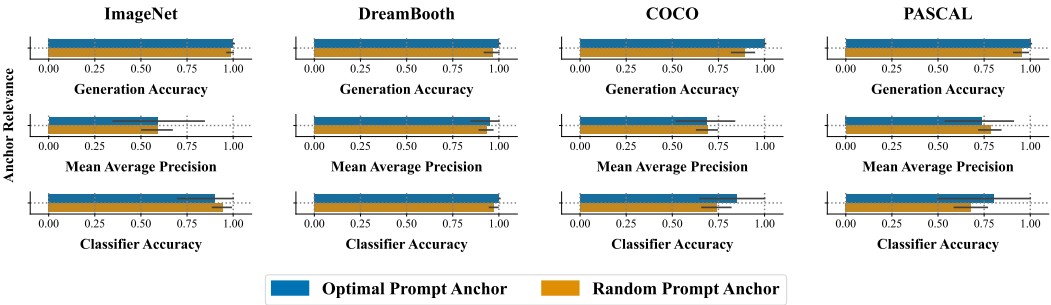

Figure 5: Performance (x-axis) of soft prompts optimized to cause generation, detection, and classification of new visual concepts under a constraint level of $\delta = 0.5$, split by the relevance of the prompt anchor (y-axis). Optimal prompt anchor corresponds to selecting the target concept name (i.e. "cat") as the anchor, and random prompt anchor corresponds to the mean performance of all other unrelated concept names (see Appendix D). Each bar corresponds to the 95% confidence interval of the corresponding metric across the 10 target concepts. We observe no impact of the relevance of the prompt anchor to the target concept in task performance.

This experiment controls the location of solutions found by prompt tuning in the embedding space, to help us understand the relationship between location, and fidelity of the solution. Using this tool, we will ask: *where are the performant solutions to prompt tuning found in the embedding space?*

**Abundance of Performant Solutions** We measure the performance of solutions from prompt tuning under varying constraint levels $\delta \in \{0.1, 0.2, 0.5, 1.0\}$, and discrete prompt anchors. Figure 4 shows a consistent behavior across models, and datasets that performance saturates at a constraint level of $\delta = 0.5$, when the nearest neighbor is still the anchor $w_{\text{anchor}}$. For constraint levels $\delta > 1$, performance does not improve further, despite the larger solution space. Solutions lose a considerable degree of performance when transferred, which suggests the perturbations $\epsilon = \vec{v}_y - y(w_{\text{anchor}})$ are model-specialized, and strikingly not emergent embeddings akin to the typical discrete prompts. Instead, performant solutions to prompt tuning are closer to adversarial perturbations than traditional prompts, and appear to be found in the vicinity of most discrete prompts in the embedding space.

**Resilience to Anchor Points** We explore the relationship between the relevance of the discrete prompt anchor, and the target concept to performance in Figure 5, and observe no major impact. We take the solutions found under a constraint level of $\delta = 0.5$, and stratify performance based on whether the discrete prompt anchor is related to the target concept. Despite intuition suggesting they would require more optimization to modify the default behavior of the model, solutions anchored to random discrete prompts perform just as well as those constrained to the optimal related anchor. The surprising resilience of prompt tuning solutions to their anchor point suggests the underlying model has such a high degree of reprogrammability that a perturbation can be found that targets any discrete prompt in the embedding space and overrides the model's original behavior for that prompt.

## 4.5 SOLUTIONS TARGET SPECIFIC LAYERS

Performant solutions are numerous in the embedding space, and these solutions are specialized. How can we tell these solutions apart from typical discrete prompts? One characteristic that identifies prompt tuning solutions is their effect on the representations predicted by the text encoder. Solutions following those in Figure 4 target the final layers of the text encoder, and steer the predicted representation towards the target concept. Figure 6 shows generations from Stable Diffusion 2.1 Rombach et al. (2022) when truncating the text encoder to just the first $N$ transformer blocks (block = Norm $\rightarrow$ Attention $\rightarrow$ Residual $\rightarrow$ Norm $\rightarrow$ MLP $\rightarrow$ Residual). The bottom row shows TSNE visualizations of the pooling token representation at four evenly spaced layers in the text encoder of Stable Diffusion when generating concepts from the ImageNet (Deng et al., 2009) task. The representations initially cluster around strawberry, the anchor concept, and generations from early layers yield strawberries instead of the target concept, sombrero. As we probe deeper layers in the text encoder, representations are gradually steered towards the target concept by the final layer. This can be observed by the evolution of TSNE clusters towards a full separation by color, and the change in generations from the initial strawberry to the final sombrero by layer 24.

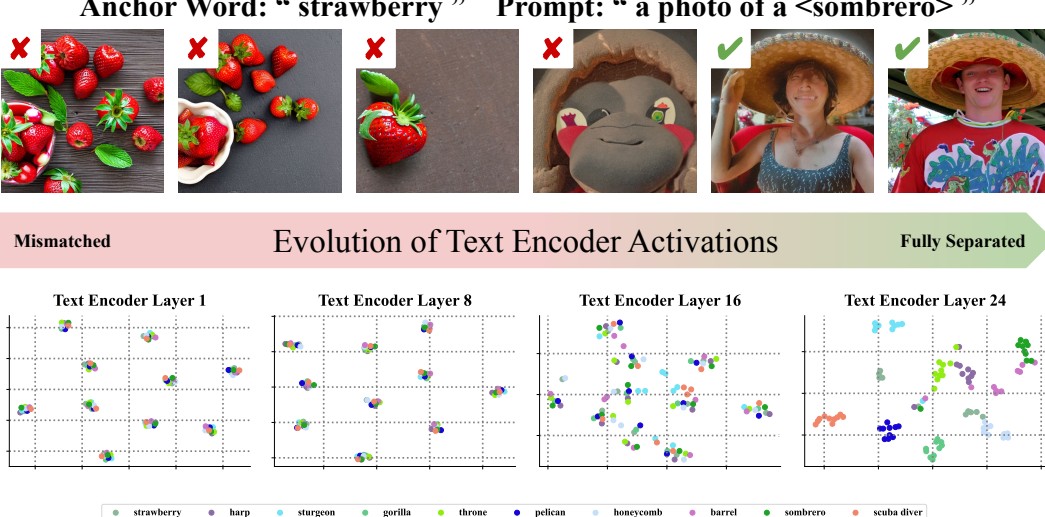

Figure 6: Prompt tuning solutions target the final layers in text encoders. We show images generated by Stable Diffusion when truncating the text encoder to the first $N$ layers, and create TSNE visualizations of the text encoder representations for the pooling token at four evenly spaced layers. Each color represents a different visual concept. Representations in plots 1-16 cluster around anchors instead of the target concept. When truncating the text encoder to just these layers, the anchor concept (strawberry) is generated instead of the target concept (sombrero). As we probe deeper into the text encoder, representations are gradually steered away from the anchor and towards the target concept so that clusters and generations are correct by layer 24. Additional visualizations in Appendix K show clusters for the remaining models and datasets.

**Understanding The Results**    The results suggest that soft prompts obtained via prompt tuning methods that encode specific visual concepts are more akin to adversarial perturbations than traditional discrete prompt embeddings. Solutions can be found in the close neighborhood of any discrete prompt that reprogram the model to generate, detect, or classify an arbitrary target concept. Prompt tuning solutions commandeer the final layers of text encoders, and steer the representations towards the target concept. The ease of finding performant solutions independent of their initialization, and anchor point suggests that Large Multimodal Models are highly reprogrammable.

## 5    SEMANTIC SIMILARITY OF INPUT EMBEDDINGS

To contextualize these results, we conduct an analysis to explore the degree of semantic similarity of the input embeddings of tested models. We seek to confirm that the input embeddings share a high degree of semantic similarity for these models, despite being training on different tasks, and to illustrate that the non-transferability of prompt-tuning solutions arises due to adversarial behavior. To begin, we will show that embeddings for discrete prompts, and randomly sampled embeddings in the neighborhood of discrete prompts are transferable between these models.

### 5.1    NON-TUNED EMBEDDINGS ARE TRANSFERABLE

To show that transferable embeddings exist, we lookup embeddings for the anchor word of each visual concept, and compare the performance of generation, detection, and classification with the anchor word embedding, to generation, detection, and classification with transferred embeddings. Importantly, when constructing the Transfer Function based on Equation 2 for this experiment, we hold our the set of anchor words as a test set. Shown in Figure 7, embeddings for anchor words are transferable in nearly every example, and nearly match the performance of the in-domain prompt. A minimal reduction in performance is observed when transferring embeddings to detection, but the resulting performance is significantly higher than transferred prompt-tuning solutions. A secondary experiment exploring the transferability of randomly sampled embeddings in the neighborhood of these discrete prompt is shown in Appendix H, and shows a similar result.

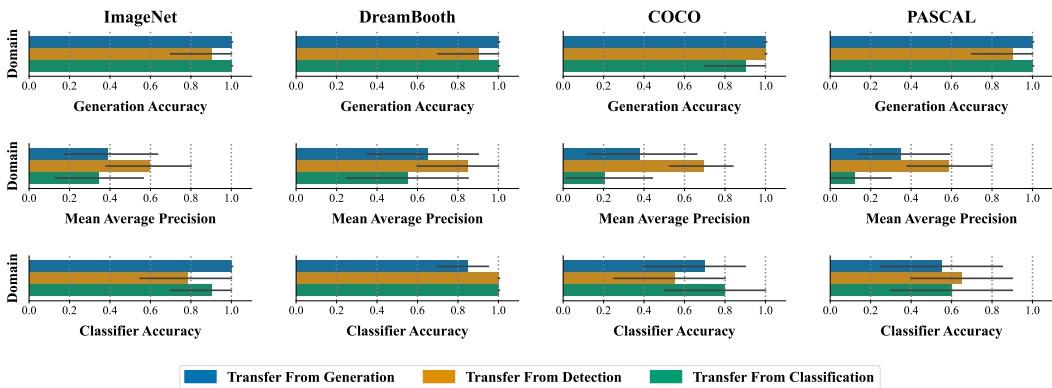

Figure 7: Transfer results for embeddings of anchor words. In nearly all cases, embeddings for existing discrete prompts can be transferred from the input embedding space of one model to another, and nearly match the in-domain performance. Base models exhibit a high degree of semantic similarity in their input embeddings.

| Task A | Task B | CKA (RBF) | Task A | Task B | CKA (Linear) |
|--------|--------|-----------|--------|--------|--------------|
| generation | detection | 0.636 | generation | detection | 0.514 |
| generation | classification | 0.759 | generation | classification | 0.633 |
| detection | classification | 0.626 | detection | classification | 0.506 |

Table 2: Values for the CKA metric between input embeddings. Using an RBF, and Linear kernel, we measure the centered kernel alignment score of the embeddings for shared tokens in the text encoders of large multimodal models. Values for CKA range from 0.5 to 0.75, which indicates a relatively high degree of semantic similarity in the structure of the input embeddings.

## 5.2 EMBEDDING SPACES ARE SIMILARLY STRUCTURED

The transferability of discrete prompts, and randomly sampled prompts in their neighborhood, suggests that large multimodal models acquire similarly structured input embedding spaces. To confirm this hypothesis, we measure the *centered kernel alignment* score (Kornblith et al., 2019) for the embeddings of shared tokens in the text encoders of these models. We report scores in Table 2. Based on ranges presented in Kornblith et al. (2019), the measured values appear relatively high, which perhaps makes the *fractured-ness* of prompt-tuning solutions more surprising than otherwise. Transferable embeddings for visual concepts do exist, but prompt tuning struggles to find them.

## 6 DISCUSSION

How do soft prompts obtained via prompt tuning methods that encode specific visual concepts (i.e. black dog) differ from typical discrete prompts? We conduct a large-scale study of prompt embeddings that encode specific visual concepts across generation, detection, and classification tasks, and show that prompt tuning solutions are akin to adversarial attacks on text encoders. Our results suggest that Large Multimodal Models have fractured embedding spaces, where perturbations can be found within an $\epsilon$-ball to any discrete prompt that reprogram the behavior of the underlying model. The ease of finding these perturbations for any discrete prompt with performance comparable to an optimal initialization suggests that Large Multimodal Models are highly reprogrammable in general. One consequence of these findings is that, without careful regularization, it is unlikely that solutions found by prompt tuning in Large Multimodal Models will be interpretable to another model.

Our work aims to galvanize the interpretability of solutions found via prompt tuning by exploring their relationship to discrete prompts, and their affect on the underlying model. We provide a benchmark for evaluating the transferability of soft prompts to enable researchers to further study and improve the interoperability of prompts following Figure 1, allowing prompts trained for generating black Labradors to be re-used for detection, and other tasks. Transferring prompts can significantly improve the adaptability and cost of machine learning systems by eliminating the need to re-train when new models are released. We highlight the difficulty of transferring soft prompts for current models, and our analysis begins to explain why transferring soft prompts is so challenging.

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

## A    LIMITATIONS & SAFEGUARDS

We employ pretrained diffusion models, object detectors, and classifiers in this work, and these models are known to have biases, obtained from their training data. Diffusion models in-particular can generate harmful or dangerous content, including graphic imagery of violence, and pornography. We employ the Stable Diffusion safety checker to flag generations after transferring soft prompts for unsafe content as a mitigation strategy for this potential limitation. Transferring soft prompts currently does not perform very well outside of certain common concepts, and one limitation of this paper is its scope: we do not propose new methodology for transferring soft prompts with high fidelity. Rather, we benchmark popular methods for soft prompt-tuning on three recent models, and show that most prompts are not transferable. Our experiments suggest that non-transferable prompts have certain properties that can be used to identify them, but turning this identification strategy into a mitigation method is outside the scope of this paper, and left for future research.

## B    ETHICAL CONSIDERATIONS

Diffusion models currently require pristine data showing a subject in clear view in order to generate new photos of that subject. Transferring soft prompts from an object detector has the potential to allow for training on less pristine data that shows the subject amidst many distracting objects. One potentially harmful consequence of transfer between object detection models and generative models is related to privacy. Individuals that don't upload photos of themselves online are currently protected from their likeness being generated by diffusion models. However, transfer from object detectors to generative models would allow for their likeness to be generated, even when photos only show them in crowded spaces. Likewise, transferring prompts from generation to detection allows for the rapid creation of detectors for specific individuals. This technology could be used by malicious actors to track the activity of specific individuals, invading their privacy.

## C    BROADER IMPACTS

Transferring prompts for specialized tasks significantly improves the adaptability and cost of machine learning systems by removing the need to re-train when new models are released. The cadence of multimodal machine learning is such that new models are released every month, and the state-of-the-art is in constant flux. Currently, soft prompts trained for older models are discarded when newer models are released, or when the task changes (i.e. classification becomes detection). Enabling the re-use of soft prompts would allow users to download prompts trained by someone else, like plugins, even when the original use-case for that soft prompt was for a different task (such as generation).

One negative broader impact that results from improved transferability is that soft prompts encoding negative and harmful behaviours become easier to use and maintain. Currently, harmful prompts become obsolete quickly as newer models are released, but once they can be transferred, they become permanent. Mitigation strategies for this risk could involve moderating online databases containing soft prompts to remove ones that perpetuate harmful behaviors, and filtering the outputs of models using the soft prompts to directly remove the harmful content (in the same vein as a safety checker).

## D    SELECTED CONCEPTS & ANCHOR WORDS

In this section, we discuss the concepts that were selected from ImageNet Deng et al. (2009), COCO Lin et al. (2014), PASCAL Everingham et al. (2010), and the DreamBooth dataset Ruiz et al. (2023). These concepts were selected uniformly at random without replacement from the available classes in each dataset. Ten classes were sampled per dataset in order to reduce the computational complexity of the experiments in the paper (results take 3 days to produce on just 40 visual concepts). These classes cover a diverse set of visual concepts.

On the ImageNet dataset Deng et al. (2009), we select [`'strawberry'`, `'harp'`, `'sturgeon'`, `'gorilla'`, `'throne'`, `'pelican'`, `'honeycomb'`, `'barrel'`, `'sombrero'`, `'scuba diver'`] as target concepts.

| Hyperparameter Name | Hyperparameter Value |
|---|---|
| Generation Model Name | Stable Diffusion 2.1 Rombach et al. (2022) |
| Generation Model HuggingFace ID | `stabilityai/stable-diffusion-2-1` |
| Generation Image Size | 768 x 768 |
| Detection Model Name | OWL-v2 Minderer et al. (2022) |
| Detection Model HuggingFace ID | `google/owlv2-base-patch16-ensemble` |
| Detection Image Size | 960 x 960 |
| Classification Model Name | Data Filtering Networks Fang et al. (2023) |
| Classification Model HuggingFace ID | `apple/DFN2B-CLIP-ViT-L-14` |
| Classification Image Size | 224 x 224 |
| Examples Per Concept | 8 |
| Embedding Vectors Per Concept | 4 |
| Denoising Steps | 50 |
| Batch Size | 8 |
| Learning Rate | 1e-04 |
| Gradient Descent Steps | 1000 |
| Optimizer | Adam |
| Adam Beta1 | 0.9 |
| Adam Beta2 | 0.999 |
| Adam Epsilon | 1e-08 |
| Weight Precision | float16 |

Table 3: Hyperparameters used in the experiments of the paper. These parameters are held constant across all datasets and models. These choices are adapted from relevant prior work.

On the DreamBooth Dataset Ruiz et al. (2023), we select [`'cat2'`, `'vase'`, `'duck_toy'`, `'candle'`, `'colorful_sneaker'`, `'backpack_dog'`, `'grey_sloth_plushie'`, `'fancy_boot'`, `'clock'`, `'pink_sunglasses'`] as target concepts.

On the COCO dataset Lin et al. (2014), we select [`'laptop'`, `'scissors'`, `'donut'`, `'bear'`, `'cup'`, `'dog'`, `'bottle'`, `'umbrella'`, `'cat'`, `'remote'`] as target concepts.

On the PASCAL VOC dataset Everingham et al. (2010), we select [`'airplane'`, `'bicycle'`, `'bird'`, `'boat'`, `'person'`, `'train'`, `'car'`, `'cat'`, `'horse'`, `'cow'`] as target concepts.

In addition to selecting concepts, we select anchor words that tokenize to a single token across all of the tested models. These are derived from the above target concepts.

On the ImageNet dataset Deng et al. (2009), we select [`'strawberry'`, `'harp'`, `'sturgeon'`, `'gorilla'`, `'throne'`, `'pelican'`, `'honeycomb'`, `'barrel'`, `'hat'`, `'scuba'`] as anchor words.

On the DreamBooth Dataset Ruiz et al. (2023), we select [`'cat'`, `'vase'`, `'duck'`, `'candle'`, `'sneaker'`, `'backpack'`, `'plush'`, `'boot'`, `'clock'`, `'sunglasses'`] as anchor words.

On the COCO dataset Lin et al. (2014), we select [`'laptop'`, `'scissors'`, `'donut'`, `'bear'`, `'cup'`, `'dog'`, `'bottle'`, `'umbrella'`, `'cat'`, `'remote'`] as anchor words.

On the PASCAL VOC dataset Everingham et al. (2010), we select [`'airplane'`, `'bicycle'`, `'bird'`, `'boat'`, `'person'`, `'train'`, `'car'`, `'cat'`, `'horse'`, `'cow'`] as anchor words.

# E  HYPERPARAMETERS

In this section, we enumerate the hyperaramters used in the experiments in the paper. We choose hyperparameters agnostic to the model and task, so that results in the experiments are general, and

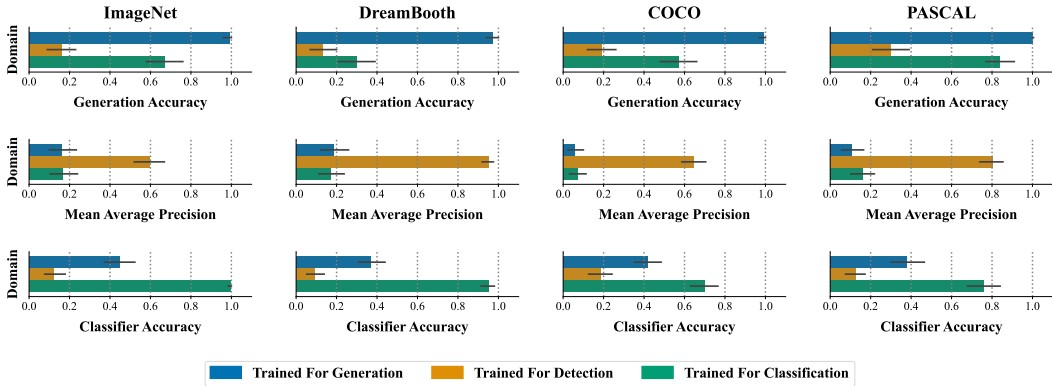

Figure 9: Visual prompt embeddings trained for one task (i.e. generation) perform well on that task, but may not perform well when transferred to another task (i.e. generation → detection). In certain directions, such as classification → generation, transfer works better than others. To understand when transfer fails, we perform extensive ablations across four standard datasets, and three models in generation, detection, and classification.

not specific to the model. In Table 3 we note the HuggingFace model ID used, model configuration details, and hyperparameters from training, and evaluation.

## F  UNCONSTRAINED SOLUTIONS

We consider a variant of the experiment in Figure 4, where we optimize prompt embedding using an unconstrained objective, and evaluate their task performance. We conduct a large-scale experiment, training 1200 new prompts for the same 40 visual concepts used in the main paper. We use standard gradient descent, instead of projected gradient descent to calculate gradients, and backpropagate gradients directly into the prompt embeddings. After training prompt embeddings to optimal performance on the training task, we evaluate on a different test task using metrics discussed in Section 4.3.

### F.1  EXPLORING TRANSFERABILITY

Results of the experiment in Figure 9 show that prompts optimized for visual tasks can perform well in-domain, but are typically not re-usable. In most transfer scenarios, prompts optimized for one task don't solve a different task than they were trained on with comparable fidelity to in-domain training. Prompts optimized for classification transfer best, achieving up to 84% of the performance of in-domain training for generation (PASCAL), and up to 28% of the in-domain detection performance (ImageNet). Prompts optimized for detection are least transferable, achieving up to 26% of in-domain classification performance (COCO), and up to 30% of in-domain generation performance (PASCAL). Prompts optimized for generation are in the middle in terms of their transferability, attaining up to 59% of the in-domain classification performance (COCO), and up to 28% of the in-domain detec-

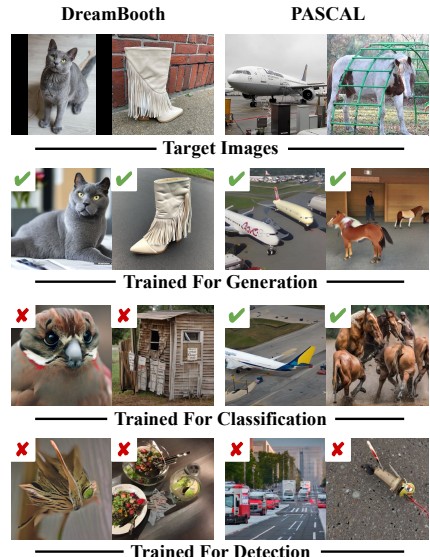

Figure 8: Generations (rows 2-4) from Stable Diffusion for target concepts (top row) from the DreamBooth and PASCAL datasets. The second row trains prompt embeddings for generation. The third row transfers prompt embeddings from classification to generation. The final row transfers from detection. prompts trained for generation capture fine-grain details. prompts trained for classification work for common concepts on PASCAL, but fail at fine-grain concepts on DreamBooth. prompts trained for detection generally don't transfer.

tion performance (ImageNet). Generation shows a significant difference in performance between prompts transferred from classification vs. detection, what's happening here?

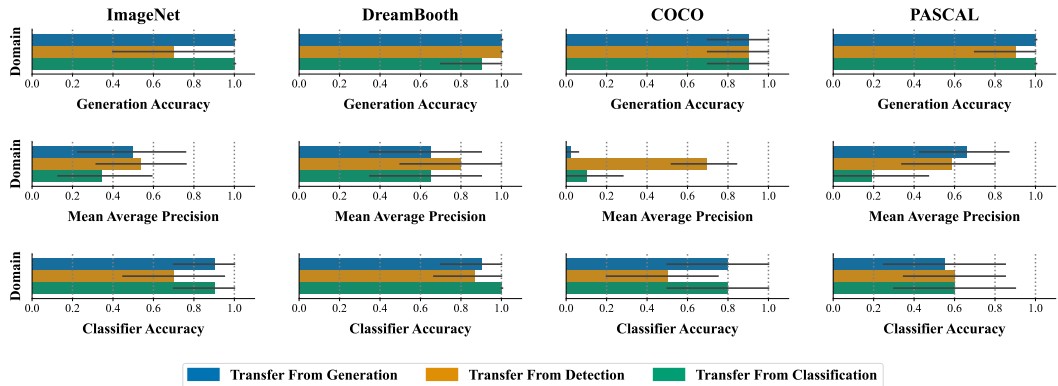

Figure 10: Transfer results for randomly sampled embeddings near anchor words. We sample an independent random perturbation for each trial from a Normal distribution with mean 0, and standard deviation 0.01, and add to the embeddings for anchor words in Figure 7. These randomly perturbed embeddings show a robustness when transferred, and nearly match the performance of the in-domain discrete prompt.

**Understanding The Results**  Using generation as a case study, we show images generated by Stable Diffusion 2.1 in Figure 8 using embeddings trained for generation (second row), transferred from classification (third row), and from detection (fourth row). We select two fine-grain concepts from the DreamBooth dataset, and two common concepts from PASCAL VOC. prompt embeddings trained for generation succeed at learning both fine-grain details for subjects in the DreamBooth dataset, and common classes in PASCAL. For prompts trained for classification, however, fine-grain details are missed, but common classes are learned. Results trained for detection miss fine-grain details, and common classes when transferred to generation, explaining patterns in Figure 9.

# G  VERIFYING THE GENERATION ACCURACY METRIC

In the table below, we report the baseline classification accuracies of ther CLIP model used in our Generation Accuracy metric. On each domain, the classifier attains an accuracy of more than 88%, suggesting that it can reliably discern each visual concept in each dataset.

| ImageNet | DreamBooth | COCO | PASCAL VOC |
|---|---|---|---|
| 98.8% | 97.5% | 88.8% | 95.0% |

Table 4: Baseline 10-way classification accuracy of the CLIP model selected for evaluating generations from Stable Diffusion 2.1, tested on each domain.

# H  TRANSFER FOR RANDOMLY-PERTURBED EMBEDDINGS

We continue the analysis from Section 5, and show that randomly sampled prompts in the neighborhood of existing discrete prompts maintain performance when transferred. We employ a Gaussian distribution with mean 0, and standard deviation 0.01 to sample perturbations, and add these perturbations to the embeddings for anchor words used in Figure 7. Results are shown in Figure 10, and illustrate that randomly sampled prompts have comparable transferability to discrete prompts. This suggests the non-transferability of prompt-tuning solutions is due primarily to adversarial behavior.

# I  MORE COMPLEX TRANSFER FUNCTIONS

In the main text of the paper, we explored a linear transfer function, and showed that under such a map, soft prompts optimized for one multimodal model were non-transferable. We provide additional results with an MLP transfer function in Figure 11, and find consistent behavior with the

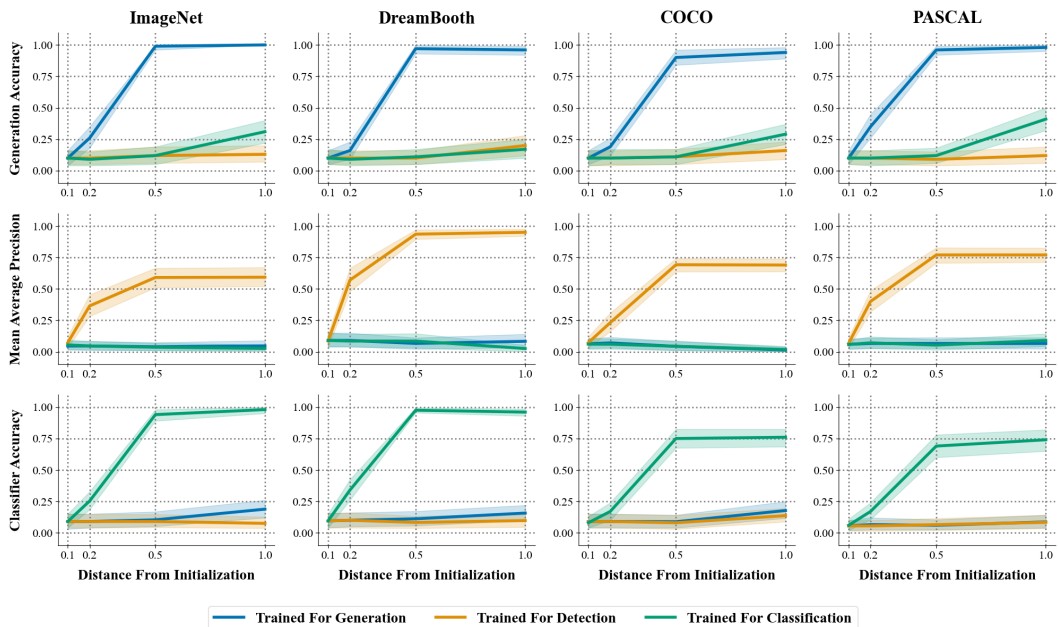

Figure 11: Performance of word vectors (y-axis) optimized for one task, and transferred to another model using a two-layer MLP transfer function, for various constraint levels $\delta$ (x-axis). Under an MLP transfer function, transferability of word vectors did not improve.

linear transfer function. We train a two-layer MLP with ReLU activation functions, and 4096 hidden units to optimize the embedding transfer loss function in Equation 1. Transfer performance of soft prompts is marginally worse with this MLP transfer function than with a linear transfer function.

## J    MORE EXAMPLES

In this section, we show more examples of generations from Stable Diffusion for perturbations to various unrelated anchor words in the embedding space. We show results for all combinations of 10 target concepts (row labels) and 10 anchor words (column labels) on ImageNet Deng et al. (2009), COCO Lin et al. (2014), PASCAL Everingham et al. (2010), and the DreamBooth dataset Ruiz et al. (2023). In several cases, nearly identical images are generated by Stable Diffusion for perturbations near to different unrelated anchor words.

## K    ADDITIONAL VISUALIZATIONS

We provide additional TSNE visualizations of the text encoder activations for different models and datasets in this section. Patterns in Section 4.5 hold across all tested models and datasets. Perturbative soft prompts like those found in Section 4.4 target the final layers in text encoders, and early activations in text encoders disagree with later activations. Generating images when truncating the text encoder to the first $N$ layers leads to generations of the anchor, instead of the target concept we are optimizing for (see Figure 6). When perturbative solutions are transferred, this transition stops.

Fine-tuning that targets the final layers of text encoders does not transfer, and Figure 22 shows that activations stop clustering by concept (color) when soft prompts are transferred.

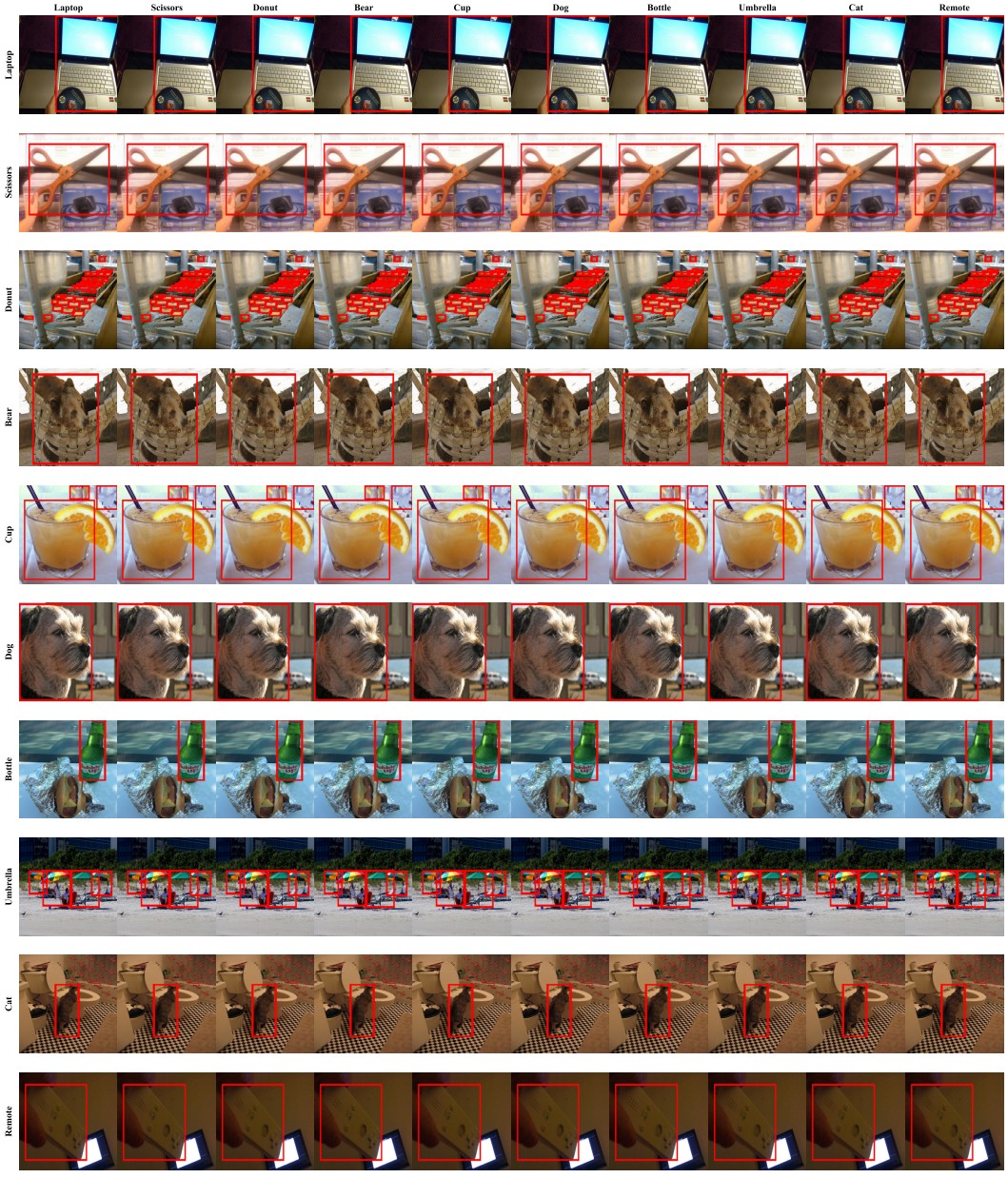

Figure 12: Visualizations of detections from OWL-v2 Minderer et al. (2022) using new embeddings optimized for detecting visual concepts on COCO Lin et al. (2014). Performant solutions for detecting arbitrary target concepts (row labels) are found with a constraint threshold $\delta = 0.5$ of unrelated anchor words (column labels).

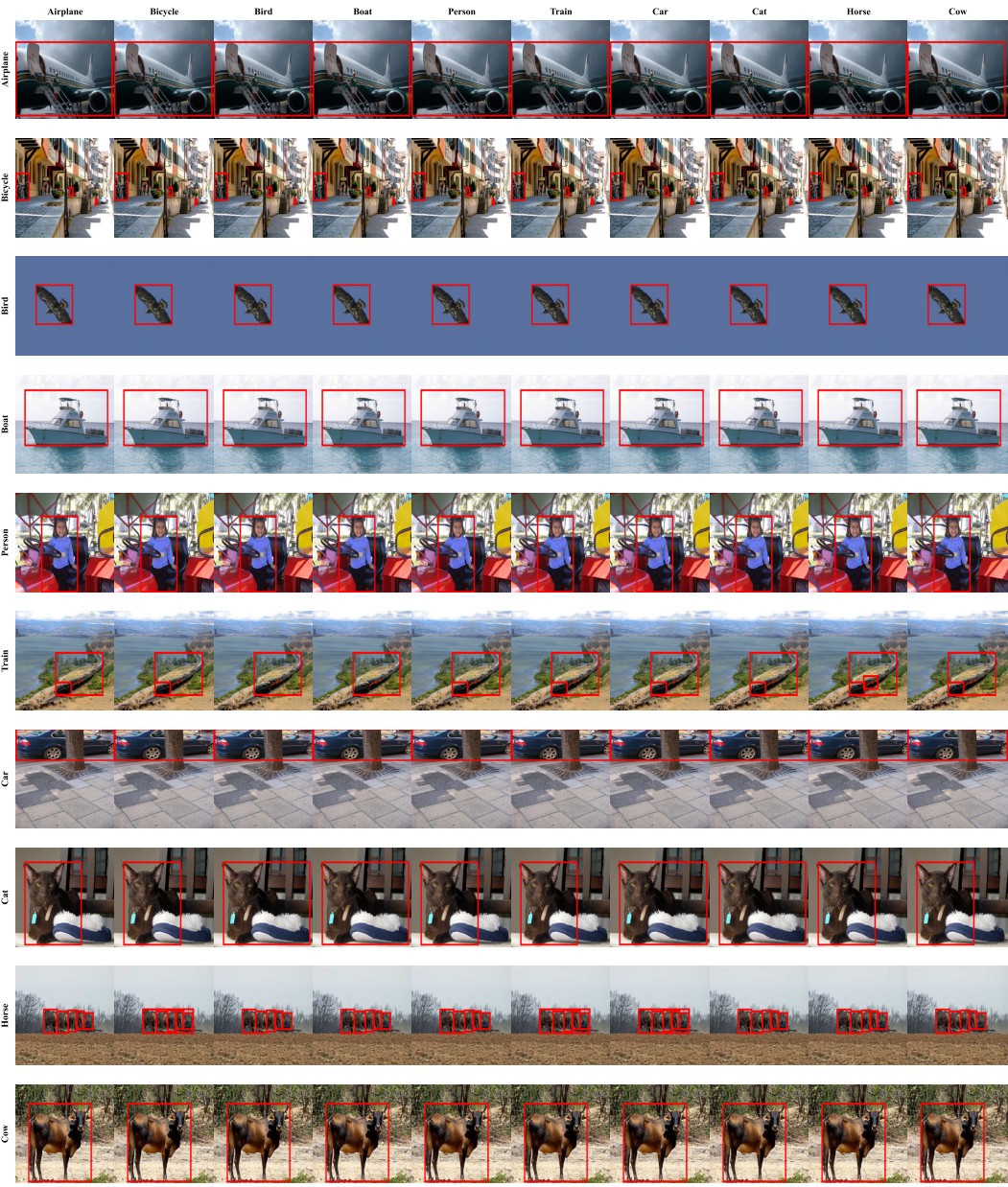

Figure 13: Visualizations of detections from OWL-v2 Minderer et al. (2022) using new embeddings optimized for detecting visual concepts on PASCAL Everingham et al. (2010). Performant solutions for detecting arbitrary target concepts (row labels) are found with a constraint threshold $\delta = 0.5$ of unrelated anchor words (column labels).

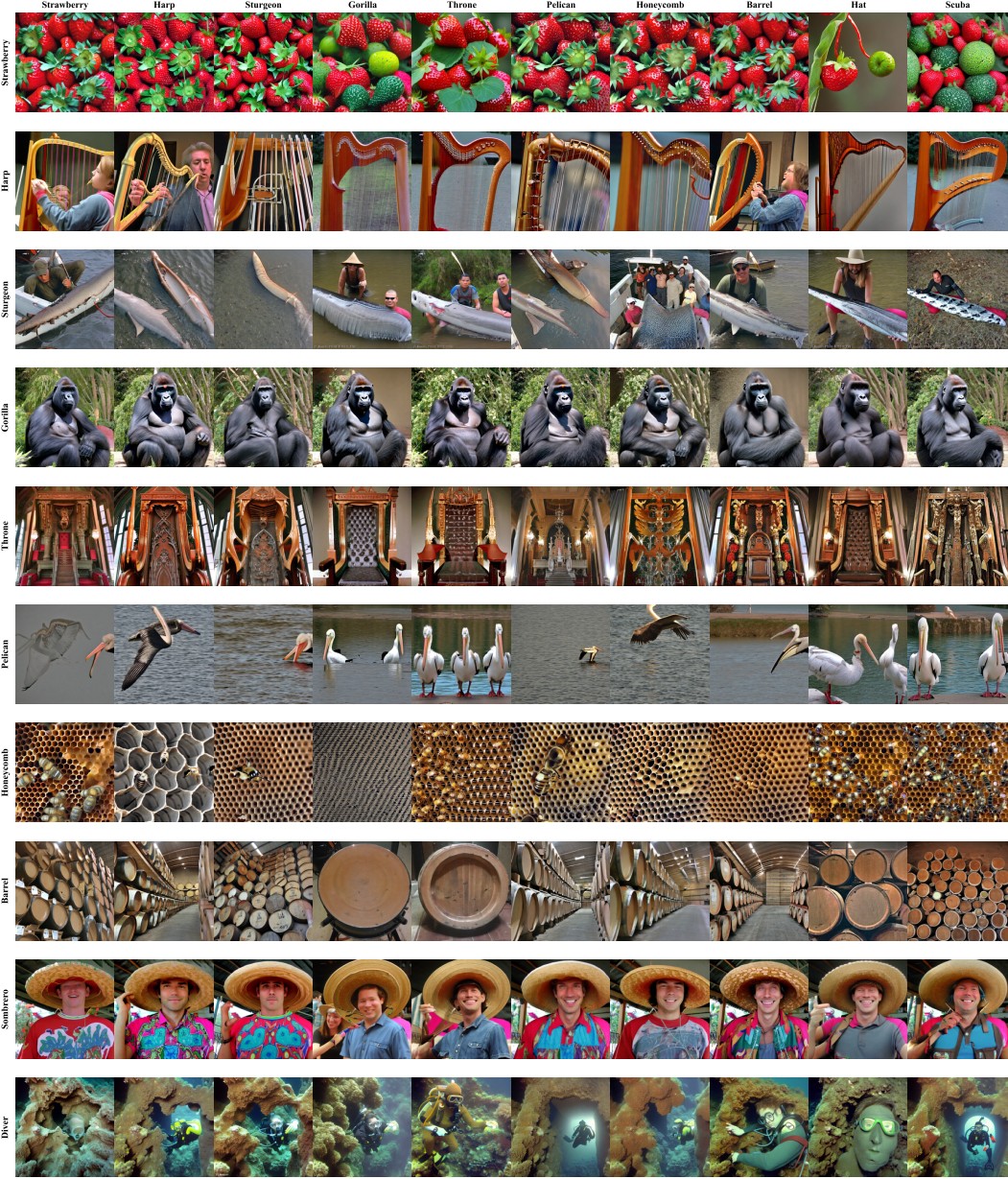

Figure 14: Visualizations of generations from Stable Diffusion 2.1 Rombach et al. (2022) using new embeddings optimized for generating visual concepts on ImageNet Deng et al. (2009). Performant solutions for generating arbitrary target concepts (row labels) are found with a constraint threshold $\delta = 0.5$ of unrelated anchor words (column labels). In several cases, different solutions far apart generate the same image.

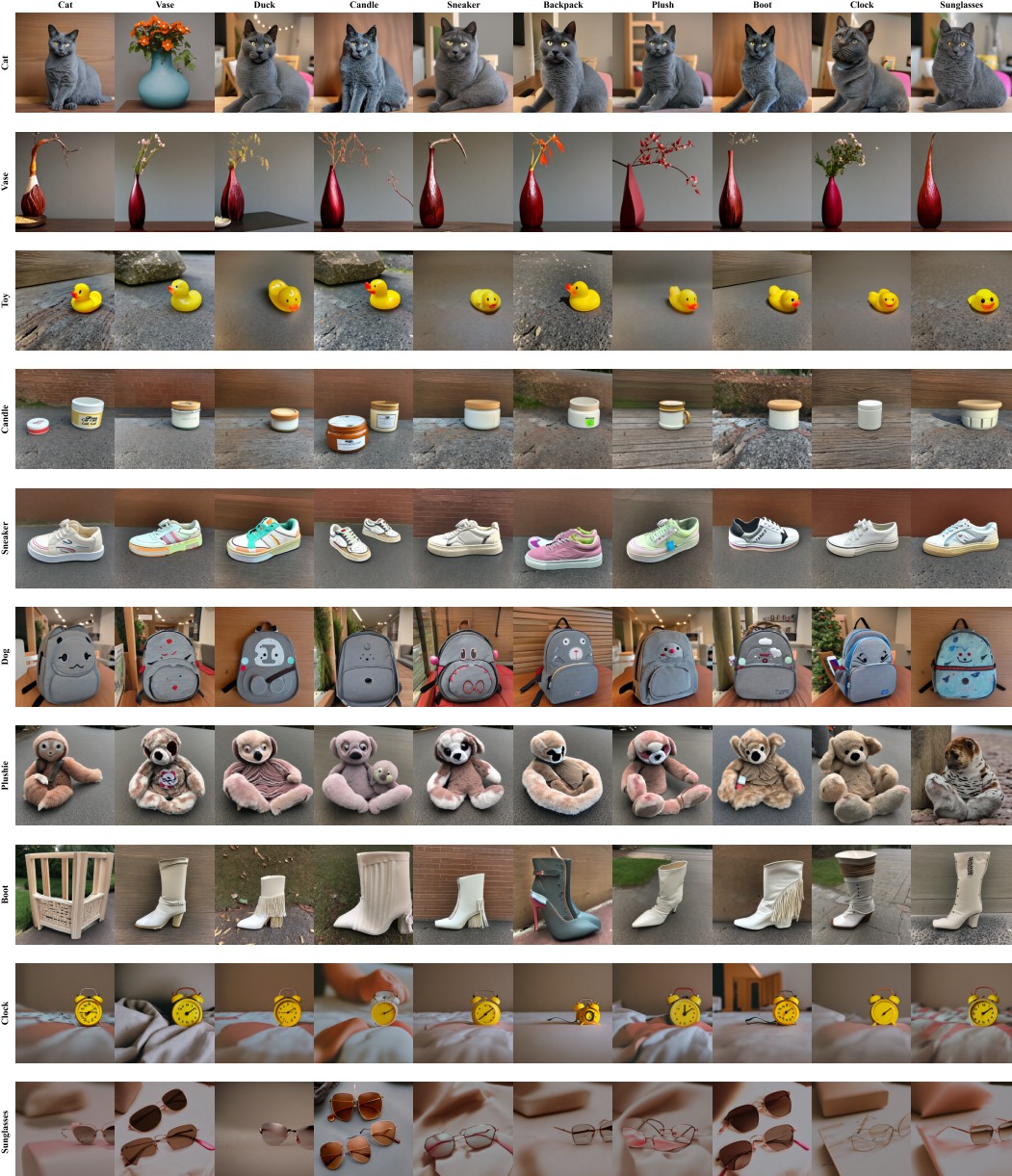

Figure 15: Visualizations of generations from Stable Diffusion 2.1 Rombach et al. (2022) using new embeddings optimized for generating visual concepts on DreamBooth Ruiz et al. (2023). Performant solutions for generating arbitrary target concepts (row labels) are found with a constraint threshold $\delta = 0.5$ of unrelated anchor words (column labels). In several cases, different solutions far apart generate the same image.

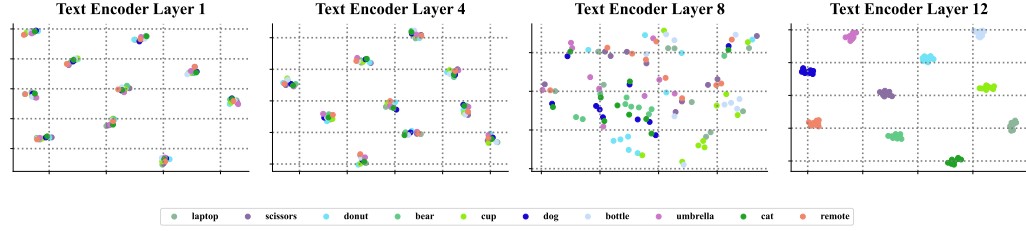

Figure 16: Visualizations of text encoder activations for OWL-v2 Minderer et al. (2022) on COCO Lin et al. (2014) at four evenly spaced layers when optimizing soft prompts for detecting visual concepts (colored points), constrained to the neighborhood of various anchor tokens (clusters in plots 1-8).

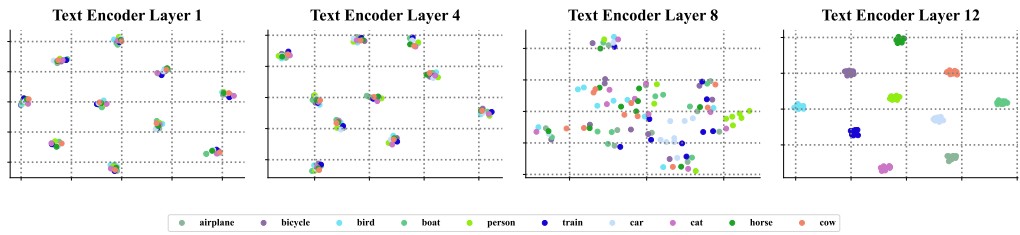

Figure 17: Visualizations of text encoder activations for OWL-v2 Minderer et al. (2022) on PASCAL Everingham et al. (2010) at four evenly spaced layers when optimizing soft prompts for detecting visual concepts (colored points), constrained to the neighborhood of various anchor tokens (clusters in plots 1-8).

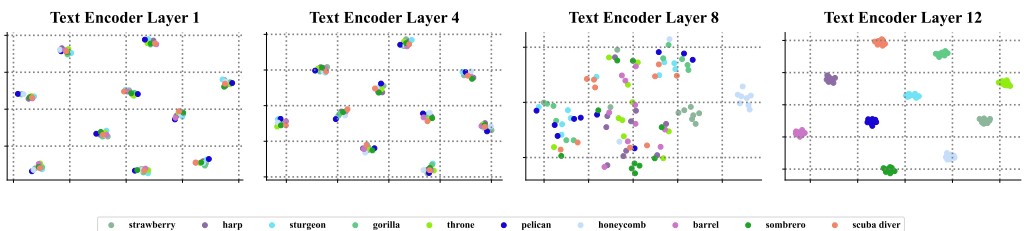

Figure 18: Visualizations of text encoder activations for DFN CLIP Fang et al. (2023); Radford et al. (2021) on ImageNet Deng et al. (2009) at four evenly spaced layers when optimizing soft prompts for classifying visual concepts (colored points), constrained to the neighborhood of various anchor tokens (clusters in plots 1-8).

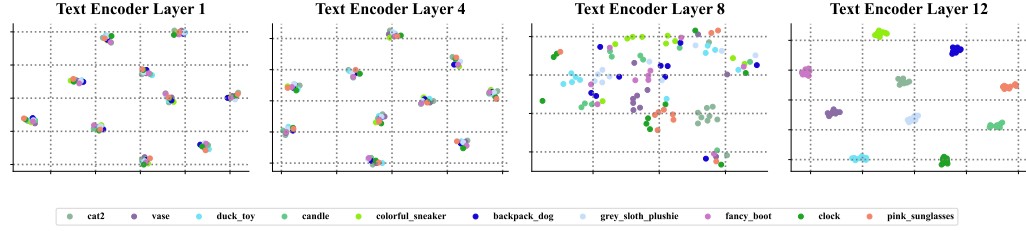

Figure 19: Visualizations of text encoder activations for DFN CLIP Fang et al. (2023); Radford et al. (2021) on DreamBooth Ruiz et al. (2023) at four evenly spaced layers when optimizing soft prompts for classifying visual concepts (colored points), constrained to the neighborhood of various anchor tokens (clusters in plots 1-8).

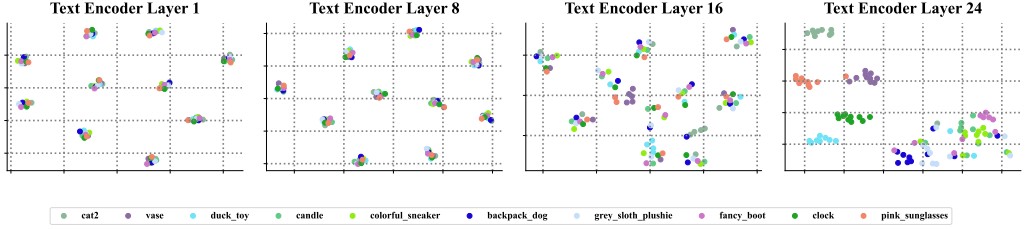

Figure 20: Visualizations of text encoder activations for Stable Diffusion 2.1 Rombach et al. (2022) on DreamBooth Ruiz et al. (2023) at four evenly spaced layers when optimizing soft prompts for generating visual concepts (colored points), constrained to the neighborhood of various anchor tokens (clusters in plots 1-16).

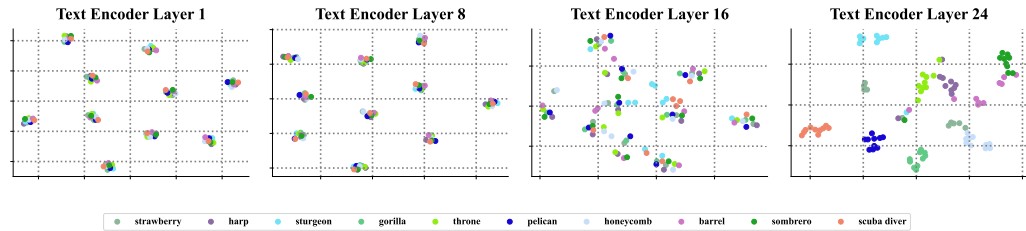

Figure 21: Visualizations of text encoder activations for Stable Diffusion 2.1 Rombach et al. (2022) on ImageNet Deng et al. (2009) at four evenly spaced layers when optimizing soft prompts for generating visual concepts (colored points), constrained to the neighborhood of various anchor tokens (clusters in plots 1-16).

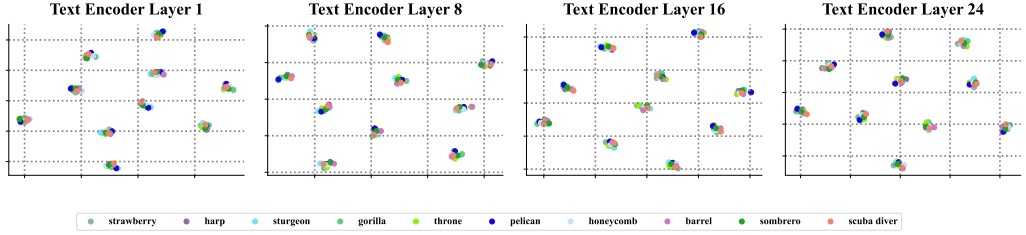

Figure 22: Visualizations of text encoder activations for Stable Diffusion 2.1 Rombach et al. (2022) on ImageNet Deng et al. (2009) at four evenly spaced layers when optimizing soft prompts for classifying visual concepts (colored points) and transferring to generation, constrained to the neighborhood of various anchor tokens (clusters in plots 1-24). The evolution of clusters towards clean separation for in-domain evaluation stops when soft prompts are transferred. Fine-tuning that targets the original model is lost.

