# OpenReview forum: "Wayward Concepts In Multimodal Models"
_ICLR.cc/2025/Conference — ICLR 2025 Poster_

### Official Review · Reviewer_1THk · 2024-10-27

**Soundness:** 3
**Presentation:** 3
**Contribution:** 2
**Rating:** 6
**Confidence:** 4

**Summary:**

The paper explores the transferability of prompt embeddings learned through prompt tuning across models trained on different tasks. It concludes that these prompt embeddings are not transferable. The study finds that prompt embeddings function similarly to adversarial perturbations, with multiple effective prompt solutions possible within close proximity to text embeddings of unrelated concepts.

**Strengths:**

1. Well-written and easy to follow.
2. Extensive analysis clearly establishes the non-transferability of learned prompt embeddings across models.
3. Perturbation analysis shows that learned embeddings constrained to random prompt anchors can perform equally well as those near related prompt anchors.

**Weaknesses:**

1. Identifies that prompt-tuned input embeddings resemble adversarial examples and lack transferability but does not propose mitigation strategies.
2. Recent works [2] [3] focuses on final-layer convergence for multimodal representations; it’s unclear if this semantic alignment exists in input embeddings. Applying metrics like CKA [1] or CKNNA [2] could reveal input embedding similarities, potentially isolating adversarial behavior as the primary non-transferability factor.
3. Lacks a control setup in the linear transformation + MSE loss analysis where the transfer works; tuning on one generative model and testing on another could serve as a useful comparison. Is the non transferability because of this Linear+MSE setup?
4. Limited transformation methods and losses explored—considers only linear transformation and MSE loss. Trying Nonlinear transformations and/or CLIP loss could offer further insights.
5. Missing citations for recent work on vision and language representation convergence, such as [3]

[1] Kornblith, Simon, et al. "Similarity of neural network representations revisited." International conference on machine learning. PMLR, 2019.
[2] Huh, Minyoung, et al. "The platonic representation hypothesis." arXiv preprint arXiv:2405.07987 (2024).
[3] Maniparambil, Mayug, et al. "Do Vision and Language Encoders Represent the World Similarly?." Proceedings of the IEEE/CVF Conference on Computer Vision and Pattern Recognition. 2024.

I am willing to raise my score if more insights are provided.

**Questions:**

1. How semantically similar (measured using CKA or CKNNA) are the input embedding representations? Are the text encoders identical across the three models? If not, do they exhibit high semantic similarity for common vocabulary but not for learned concepts?

2. Why was only MSE loss considered for learning transformations across modalities? Is the failure due to difficulty of transferring between tasks, or could it be attributed to the limitations of using a linear transform or the MSE loss?

3. What happens when the prompt is learnt for one generative model and transferred to another generative model?

---

> ### Author Response · Authors · 2024-11-22
> **Response To Reviewer 1THk 1/4**
>
> Thank you for your detailed review, compliments on aspects of the paper, and suggestions for improving the manuscript. Based on your feedback, we have conducted several new experiments in this rebuttal, and we provide discussion and clarifications to the points made in your review.
>
> There were several points made in the review, including: (1) Are base embeddings semantically similar, and can we isolate adversarial behavior as the primary non-transferability factor, (2) Providing a control experiment to verify the Transfer Function is an effective map, and (3) Exploring more transfer methods.
>
> We conduct new experiments and ablations that address these points:
>
> ## Response Summary
>
> * **Addressing Point (1)**: We conduct an experiment using the Mutual Nearest Neighbors metric (Mnn) proposed in “The Platonic Representation Hypothesis” by Hut et al. 2024. Using the same hyperparameters as their work to ensure that values are directly comparable, we find a semantic similarity between `0.157 - 0.215`, which *exceeds the semantic similarity of Dinov2 and Llama3 from Hut et al. 2024*, suggesting a high similarity.
>
> * **Addressing Point (2)**: We have added a control experiment to the paper, showing a regime where the transfer function successfully maps performant solutions between two spaces. Transferable vector embeddings exist for all tested models, suggesting the non-transferability property is not due to the Transfer Function.
>
> * **Addressing Point (3)**: We have conducted a series of ablations on the transfer function, including sparsity regularization based on an L1 penalty on the linear transformation matrix T, and a different loss function, replacing the original L2 loss with the L1 loss. Findings are not impacted by these changes.
>
> Additional discussion for these points is provided below.

---

> ### Author Response · Authors · 2024-11-22
> **Response To Reviewer 1THk 2/4**
>
> ## Base Embeddings are Semantically Similar
>
> The first point in the review pertains to whether the base embedding spaces are semantically similar. We agree with the reviewer that showing their similarity is important, as a high similarity would make our results more surprising---that models with similarly structured embedding spaces have incompatible prompt tuning solutions.
>
> * **High Semantic Similarity According to Mutual Nearest Neighbors**
>
> To approach this experiment, we adapt the Mutual Nearest Neighbors metric (Mnn) from “The Platonic Representation Hypothesis” by Hut et al. 2024 [1]. To ensure that our measured value for this metric is directly comparable to results reported in the original paper from Hut et al. 2024, we employ their hyperparameters.
>
> | Task A     | Task B         |      Mnn |
> |:-----------|:---------------|---------:|
> | generation | detection      | 0.21537  |
> | generation | classification | 0.164991 |
> | detection  | classification | 0.157687 |
>
> The baseline similarity between Dinov2 and Llama3 is `0.16`, and values for the models we tested are as high or higher than this baseline, **suggesting base embeddings for all tested models are semantically similar.**
>
> * **Considering An Alternative Metric**
>
> For completeness, we note a potential limitation of the Mnn metric---euclidean distance is perhaps more useful for general representations than for input embeddings, which are often spherically distributed. Cosine similarity may be more suitable to compare input embedding spaces than euclidean distance, so we re-compute the Mnn metric using cosine similarity instead of euclidean distance in the following table.
>
> | Task A     | Task B         |      Mnn |
> |:-----------|:---------------|---------:|
> | generation | detection      | 0.350069 |
> | generation | classification | 0.35019  |
> | detection  | classification | 0.320711 |
>
> Results show that for the k = 10 nearest neighbors according to the cosine similarity, on average roughly `35%` of the neighboring tokens are the same across all three classes of models.
>
> * **Interpreting The Findings**
>
> Given that all tested models have embedding spaces with similar structure, it is perhaps more surprising that models with similarly structured embedding spaces have incompatible prompt tuning solutions.
>
> [1] The Platonic Representation Hypothesis, Huh, Minyoung, et al., ArXiv 2024.

---

> > ### Comment · Reviewer_1THk · 2024-11-25
> > **Semantic Similarity between Embedding Spaces is still not clear**
> >
> > Dear Authors,
> > Thank you for the clarifying experiments. I have a few questions.
> >
> > 1. Which dataset was used for measuring the semantic similarity? It would be good to make this clear, as the CKA/ CKNNA scores are usually sensitive to the dataset.
> > 2. Semantic similarity of llama to dinov2 might not be a good baseline to compare against -- because semantic similarity metrics are not universal. What are the semantic similarity scores of different input embeddings when compared to dinov2? Are they lower or higher than 0.16 (dinov2--llama3)? Platonic Representations compare against dinov2 to show that language embeddings converge to that of a strong visual representation. I believe the scores are only comparable when compared against dinov2. Since the embeddings you're comparing here are from the same modality, I am unsure if the CKNNA score of a cross-modal pair (dinov2-llama) as the baseline is enough.
> > 3. What are the semantic similarity scores between the final embedding layers of the same 3 model pairs in the table shown above?

---

> > > ### Author Response · Authors · 2024-11-25
> > > **On Semantic Similarity 1/3**
> > >
> > > Thank you for following up on our rebuttal, we answer your questions below:
> > >
> > > ## Clarifying The Dataset & Steps
> > >
> > > One encounters a challenge when using similarity metrics like CKNNA and Mnn to compare input embeddings. Input embeddings are a weight of the underlying model, and not a function of a dataset, which is different from model activations, which require applying weights to a dataset to obtain representations of that dataset.
> > >
> > > With this in mind, similarity metrics must be adjusted to compare input embeddings.
> > >
> > > **We provide the steps we used to compute Mnn below:**
> > >
> > > 1. **Find Shared Tokens**: The tested models employ *slightly* different runs of the byte pair tokenization algorithm, so we must first take the intersection of the three tokenizers to find shared tokens. We record the proportion of tokens in this intersection, and `86%` of tokens from each model (`35,271`) remain after this step.
> > >
> > > The set of `35,271` shared tokens corresponds to the *“dataset”* in our case.
> > >
> > > 2. **Lookup Token Embeddings**: We then take each token kept by step 1, and we lookup the embedding corresponding to that token in the input embeddings of the base model, for all three models.
> > >
> > > 3. **Compute Similarity Metrics**: After step 2, we have three sets of token embeddings, one for each model we aim to test, and each set contains embeddings for a shared set of tokens. We select two pairs of sets, and compute semantic similarity metrics following Platonic Representations [1], using `k = 10` nearest neighbors.
> > >
> > > We are happy to elaborate more on these steps.

---

> > > > ### Author Response · Authors · 2024-11-25
> > > > **On Semantic Similarity 2/3**
> > > >
> > > > ## On Embedding Similarity
> > > >
> > > > The new question raised pertains to how we can discern if the semantic similarity measured by steps 1-3 implies a high or low similarity. We agree that Dinov2 against Llama3 may not be the most informative baseline. We appreciate the ideas for better baselines to answer this question, but the adjustments required to apply similarity metrics to model weights (i.e. input embeddings) make the requested comparisons tricky.
> > > >
> > > > **The Control Experiment Implies High Similarity**: In the control experiment, we show discrete prompts, and randomly sampled prompts in their neighborhood, are linearly transferable. The existence of a linear map that attains high transfer performance in control experiments implies semantically similar input embeddings.
> > > >
> > > > **Findings Don’t Rely On Similarity**: The goal of the paper is to understand how solutions found via prompt tuning differ from traditional discrete prompts, and our work highlights two key ways in which they differ:
> > > >
> > > > * **(Property 1)**: Models have fractured embedding spaces with many prompt tuning solutions that attain the same performance in different locations, and these prompt tuning solutions are non-transferable, despite the existence of linearly transferable solutions based on the results from the control experiments.
> > > >
> > > > * **(Property 2)**: Prompt tuning solutions target the final layers in models.
> > > >
> > > > Neither of these properties requires the precise semantic similarity value of the base input embeddings. Understanding this similarity is primarily helpful for contextualizing the paper---it becomes more surprising that prompt tuning solutions are non-transferable the more similar in structure the input embeddings become.

---

> > > > > ### Comment · Reviewer_1THk · 2024-11-27
> > > > > **Details on the control Experiment**
> > > > >
> > > > > Dear Authors,
> > > > >
> > > > > Thank you for providing clarifying answers to my questions. I have a couple more questions regarding the control experiment and semantic similarity between the embedding spaces.
> > > > >
> > > > > 1. You say that the transfer function is learnt on > 40,000 common words. Does this include the discrete tokens you successfully transfer in the control experiment? Does the transfer function work equally well on discrete tokens when trained on a set of tokens and evaluated on a different set of tokens?
> > > > >
> > > > > 2.  Could you please provide CKA scores between the different input embeddings ?
> > > > >
> > > > > Thank you

---

> ### Author Response · Authors · 2024-11-22
> **Response To Reviewer 1THk 3/4**
>
> ## Control Experiment for the Transfer Function
>
> A second point in the review pertains to the effectiveness of the Transfer Function. To address this question, we conduct an experiment showing cases where transfer succeeds, and transferred embeddings attain high performance, proving these solutions exist, but that prompt tuning does not recover these solutions.
>
> We show two successful transfer scenarios:
>
> * **Transferring discrete prompts**
>
> In this experiment, we take the embeddings for tokens of the class name for the target consent (i.e. “sombrero” for the sombrero class), and we transfer embeddings between models following the methodology in Section 4. Results for this experiment can be viewed at the anonymous link below:
>
> [Discrete prompts transfer results](https://drive.google.com/file/d/1yP7_DTPpJaos195l5bSQdOZ6plZDGRsl/view?usp=sharing)
>
> Transfer succeeds in all cases, suggesting the Transfer Function is an effective map.
>
> * **Transferring sampled prompts**
>
> Following up the previous experiment, we conduct a second experiment where we sample embeddings in the neighborhood of embeddings for tokens of the class name for the target concept. In particular, we employ a normal distribution centered at the embedding for tokens of class names, with a standard deviation proportional to the distance between tokens and their closest neighbor (so samples stay in their original neighborhood).
>
> Results for this experiment can be viewed at the anonymous link below:
>
> [Sampled prompts transfer results](https://drive.google.com/file/d/1JQNhSSLdzC96cChSrjoMWUV0qI5rr58V/view?usp=sharing)
>
> Transfer succeeds in nearly all cases, confirming that transferable solutions exist beyond discrete prompts.
>
> ## Exploring Different Transfer Functions
>
> Findings in Section 4 are robust to the design of the Transfer Function. We illustrate this by making two modifications to the original Transfer Function, which minimized a least squares loss (Equation 2).
>
> * **Modification 1 - Changing to L1 loss**
>
> Based on your suggestion, we reproduced the results in Figure 4 of Section 4 after replacing the original least squares loss with an L1 loss instead. The objective for this new Transfer Function is:
>
> $\arg \min_{T} \; \mathbb{E} \left\| \vec{x}(w) - T \vec{y}(w) \right\|_1$
>
> Results for this ablation can be viewed here:
>
> [L1 loss](https://drive.google.com/file/d/1vKf97_79Wsqi5OYbiuOZwyzxIYj85Dm9/view?usp=sharing)
>
> For all tested models and datasets, results agree with original findings.
>
> * **Modification 2 - Adding sparse regularization**
>
> For completeness, we conduct a second ablation using a sparse regularization term that penalizes the L1 norm of the linear transformation matrix T added to the original L2 loss. Specifically, the objective is:
>
> $\arg \min_{T} \; \mathbb{E} \left\| \vec{x}(w) - T \vec{y}(w) \right\|^2_2 + \lambda \left\| T \right\|_1$
>
> Results for this ablation can be viewed here:
>
> [Sparse regularization](https://drive.google.com/file/d/1R7a9gm5jYAtTuQ6x6K90kiRng0P6UiJ4/view?usp=sharing)
>
> Findings in both ablations agree with the original findings, suggesting that **conclusions drawn in our study are not impacted by the Transfer Function,** and are deeper properties of the underlying models.
>
> * **Nonlinear transfer functions**
>
> We also highlight *Appendix H, Figure 9* using a two-layer MLP Transfer function. Results in this ablation are consistent with the two modifications provided above, and reinforce the existing message of Section 4:
>
> *(Point A)* Prompt tuning finds fractured solutions.
>
> *(Point B)* One property of these solutions is they are non-transferable.
>
> *(Point C)* Another property of fractured solutions is they target specific layers in the models.
>
> We believe the consistency of the findings when varying the transfer method, and verifying the effectiveness of the Transfer Function as a map between the two spaces can help improve your confidence in our study.

---

> ### Author Response · Authors · 2024-11-22
> **Response To Reviewer 1THk 4/4**
>
> ## Miscellaneous Points
>
> **”does not propose mitigation strategies”**
>
> The goal of this work is to understand how solutions found via prompt tuning methods differ from traditional discrete prompts, and the non-transferability problem we discover is one result of our pursuit of this question. We also identify that prompt tuning solutions have a second property: they target the final layers in models.
>
> We believe the new experiments added in this rebuttal show that even models with similarly structured embedding spaces have incompatible prompt tuning solutions. This highlights the importance of a paper dedicated solely to understanding **how prompt tuning solutions differ from discrete prompts.**
>
> **“Are the text encoders identical across the three models?”**
>
> No, all models have different text encoders with different weights. Based on previous experiments, their embeddings share a high degree of semantic similarity according to the Mutual Nearest Neighbors metric.
>
> **”Missing citations for recent work on vision and language representation convergence”**
>
> We are adding these citations to the manuscript, thank you for the recommendations.
>
> **”What happens when the prompt is learnt for one generative model and transferred to another generative model?”**
>
> In a new experiment, we evaluate the transferability of embeddings from Stable Diffusion 2.1 (SD21) to Stable Diffusion 1.5 (SD15), two models of the same class, but of different sizes and different weights.
>
> [Results for transfer from SD21 to SD15](https://drive.google.com/file/d/1Yl97X-ciqmwdDCpkb9eUzX8DBFeBuOd_/view?usp=sharing)
>
> Findings are consistent with existing experiments in Section 4, and show that even two models of the same class (generation in this case) suffer from the fractured property.

---

> ### Author Response · Authors · 2024-11-25
> **On Semantic Similarity 3/3**
>
> ## Understanding Similarity Via Corruptions
>
> With the additional context on the subtlety of input embedding similarity, we conduct an experiment to understand if the measured Mnn values are relatively high. We conduct a perturbation analysis that controls the degree of similarity via random corruptions.
>
> **Analysis On Random Corruptions**: In this ablation, we take input embeddings from Stable Diffusion 2.1 following steps 1-3, and measure Mnn values with respect to a randomly corrupted version of the embeddings. For a *Corruption Strength* between 0% and 100%, we replace a random subset of that many token embeddings with random vectors from a unit normal distribution, and compute the Mnn similarity between the original and corrupted versions.
>
> Results are shown below for euclidean and cosine similarity metrics:
>
> * **Mnn With The Euclidean Distance Metric**
>
> | name                |   0 |        1 |        2 |        3 |       4 |        5 |        6 |        7 |        8 |         9 |          10 |
> |:--------------------|----:|---------:|---------:|---------:|--------:|---------:|---------:|---------:|---------:|----------:|------------:|
> | Corruption Strength |   0 | 0.1      | 0.2      | 0.3      | 0.4     | 0.5      | 0.6      | 0.7      | 0.8      | 0.9       | 1           |
> | Mnn                 |   1 | 0.86 | 0.75 | 0.64 | 0.54 | 0.43 | 0.31 | 0.20 | 0.10 | 0.02 | 0.00 |
>
> * **Mnn With The  Cosine Similarity Metric**
>
> | name                |   0 |        1 |        2 |        3 |        4 |        5 |        6 |        7 |         8 |         9 |          10 |
> |:--------------------|----:|---------:|---------:|---------:|---------:|---------:|---------:|---------:|----------:|----------:|------------:|
> | Corruption Strength |   0 | 0.1      | 0.2      | 0.3      | 0.4      | 0.5      | 0.6      | 0.7      | 0.8       | 0.9       | 1           |
> | Mnn                 |   1 | 0.83 | 0.68 | 0.55 | 0.42 | 0.31 | 0.20 | 0.12 | 0.057 | 0.015 | 0.00 |
>
> The ablation reveals that an Mnn value of `0.2` for the euclidean distance metric is congruent to a regime where 30% of token embeddings are the same between the original and corrupted versions. Similarly, an Mnn value of `0.35` for the cosine similarity metric is congruent to a regime where between 50% and 60% of token embeddings are the same.

---

> ### Author Response · Authors · 2024-11-30
> **Clarification for Control Experiment & Embedding Similarity**
>
> Dear Reviewer 1THk, thank you for your feedback and engagement with our work. We address new questions and provide experiments that enforce a train-test split for the transfer function, and measure CKA values. Results show: (1) the transfer function is robust, and (2) CKA values appear relatively high.
>
> ## Answers To Questions
>
> **(Question 1)**: The transfer function successfully maps held-out prompts that were not observed during training. We provide a revised control experiment that excludes the discrete prompts we successfully transfer from the dataset used to optimize the transfer function. Results for the revised control experiment are provided below.
>
> [Control experiment on held-out discrete prompts](https://drive.google.com/file/d/1IZZ1fc2doLHxBI1npP8C-zQlOuPYNuPm/view?usp=sharing)
>
> The revised control experiment is *nearly identical* to the original, and transfer succeeds in all cases where it had originally succeeded, suggesting the transfer function is robust. Kornblith, et al. [2] discuss the utility of linear regression for comparing the structural similarity of neural representations, and conclude that a performant linear map implies a high similarity between two spaces. The success of the transfer function in this control experiment implies the input embeddings have a relatively high structural similarity.
>
> [2] Similarity of Neural Network Representations Revisited, Kornblith, Simon, et al., ICML 2019.
>
> **(Question 2)**: We provide CKA scores between all pairs of domains below, using a linear kernel, and an RBF kernel with bandwidth $\sigma$ equal to `0.8` times the mean distance between embeddings---a comparable $\sigma$ to experiments in [2]. CKA scores range from `0.5` to `0.75`, which indicates a relatively high similarity compared to scores in Figures 2-5 from [2].
>
> * **CKA Scores for an RBF Kernel**
>
> | Task A     | Task B         |      cka |
> |:-----------|:---------------|---------:|
> | generation | detection      | 0.635822 |
> | generation | classification | 0.759368 |
> | detection  | classification | 0.626023 |
>
> * **CKA Scores for a Linear Kernel**
>
> | Task A     | Task B         |      cka |
> |:-----------|:---------------|---------:|
> | generation | detection      | 0.514325 |
> | generation | classification | 0.633389 |
> | detection  | classification | 0.506291 |
>
> ## Similarity Is Understood Three Ways
>
> Our rebuttal considers the structural similarity of input embeddings via *three parallel analyses*, and all three analyses point towards the similarity being relatively high. We first explore Mnn scores using a dataset of shared tokens, and conduct a perturbation analysis that reveals scores are congruent to `30%` to `60%` of embeddings matching. Second, we provide a control experiment that confirms high transfer function performance, and implies similar embeddings based on discussion in [2]. Finally, we compute CKA scores, which range from `0.5` to `0.75`, and suggest a high similarity based on the values from Figures 2-5 in Kornblith, et al. [2].
>
> The similarity between input embeddings provides important context for our paper, and shows that prompt tuning finds incompatible solutions even for models with similar embeddings, suggesting that adversarial behavior is the primary culprit for the non-transferability of prompt tuning solutions.
>
> ## The Deciding Vote
>
> We now present stronger evidence for structural similarity of the input embeddings, which reinforces the validity and importance of our findings. The other reviewers have raised their scores to accept. With your vote serving as the deciding factor, we hope these updates provide clarity and demonstrate the significance of findings in the paper.
>
> Best, The Authors

---

> > ### Comment · Reviewer_1THk · 2024-12-02
> >
> > Dear Authors,
> >
> > Thank you for the additional experiments. It is a surprising finding that a high similarity exists between the input embeddings of different text models, and it would be good to point this out in the paper, along with the control experiment where the transfer succeed. This would make it clear that the lack of transferability is primarily due to the adversarial nature of the solutions found.
> >
> > I find the responses satisfactory and would like to raise my rating to 6.
> >
> > Thanks

---

### Official Review · Reviewer_2W9n · 2024-10-28

**Soundness:** 2
**Presentation:** 3
**Contribution:** 3
**Rating:** 6
**Confidence:** 3

**Summary:**

In this paper, the authors are motivated to learn prompts or tune prompts across different vision tasks. They mostly try to learn prompts for generative tasks and then transfer these learned prompts to discriminative tasks such as classification and detection.

**Strengths:**

1. The idea has practical significance and learning soft prompts that can transfer for various tasks is valuable.
2. The proposed linear layer learned is simple to understand and the implementation details are properly explained.

**Weaknesses:**

1. I question the value of this setup if both the generative and discriminative VLMs use the same text encoder. The authors chose Stable Diffusion 2, which uses OpenCLIP L14 for generation, and for classification, they use CLIP L14. However, if OpenCLIP L14 were also used for discriminative tasks, the problem setup might not hold.
2. Additionally, the paper lacks a discussion on the limitations of their problem setup and proposed methods, which would be helpful for understanding the approach's constraints.
3. The vocabulary in some sections feels unnecessarily complex and could be simplified for better clarity.

**Questions:**

1. What if you were to reverse the setup, learn prompts for discriminative tasks and transfer to generative tasks would the results hold?
2. Did you finetune any layers of the models? To the best of my knowledge, they seemed to have been left frozen.

---

> ### Author Response · Authors · 2024-11-23
> **Response To Reviewer 2W9n 1/4**
>
> Thank you for reviewing our paper and providing feedback on the manuscript, there are several points discussed in the review that we address in this rebuttal: (1) Clarifying our goal in this paper, (2) Discussing why the transfer setting explored in this paper is important towards our goal, and (3) Limitations of the study.
>
> We conduct new experiments and ablations that address these points:
>
> ## Response Summary
>
> * **Addressing Point (1)**: We think our goal may be misunderstood. Our goal is to understand how solutions found via prompt tuning methods differ from traditional discrete prompts, and the transferability experiments we conduct serve to reveal a key manner in which they differ. In the following rebuttal, we include new experiments that show the input embeddings of tested models exhibit high semantic similarity in their local structure, and even models with similarly structured embedding spaces have incompatible prompt tuning solutions.
>
> This highlights the importance of a paper dedicated to understanding prompt tuning solutions.
>
> * **Addressing Point (2)**: Transfer is an important component of this study because it highlights a surprising property of prompt tuning solutions: these solutions are model-specific, whereas new experiments show that many non-optimized embeddings are linearly transferable, and maintain high performance.
>
> *It suggests that adversarial behavior is the primary factor impacting transferability.*
>
> * **Addressing Point (3)**: One limitation of this study is the breadth of different Transfer Functions explored. We now address this limitation and conduct a series of ablations on the transfer function, including sparsity regularization based on an L1 penalty on the linear transformation matrix T, and a different loss function, replacing the original L2 loss with the L1 loss. Findings are not impacted by these changes.
>
> Additional discussion for these points is provided below.

---

> > ### Author Response · Authors · 2024-11-23
> > **Response To Reviewer 2W9n 3/4**
> >
> > ## Transferability Reveals a Surprising Property
> >
> > Our transfer setup is valuable because it *reveals a hidden property of prompt tuning solutions* that would be harder to identify if models shared the same weights. To show that prompt tuning solutions are unique in this aspect, we conduct an experiment showing cases where transfer succeeds, and transferred embeddings attain high performance, proving these solutions exist, but that prompt tuning does not recover these solutions.
> >
> > We show two successful linearly transferable scenarios:
> >
> > * **Transferring discrete prompts**
> >
> > In this experiment, we take the embeddings for tokens of the class name for the target consent (i.e. “sombrero” for the sombrero class), and we transfer embeddings between models following the methodology in Section 4. Results for this experiment can be viewed at the anonymous link below:
> >
> > [Discrete prompts transfer results](https://drive.google.com/file/d/1yP7_DTPpJaos195l5bSQdOZ6plZDGRsl/view?usp=sharing)
> >
> > Transfer succeeds in all cases, suggesting the Transfer Function is an effective map.
> >
> > * **Transferring sampled prompts**
> >
> > Following up the previous experiment, we conduct a second experiment where we sample embeddings in the neighborhood of embeddings for tokens of the class name for the target consent. In particular, we employ a normal distribution centered at the embedding for tokens of class names, with a standard deviation proportional to the distance between tokens and their closest neighbor (so samples stay in their original neighborhood).
> >
> > Results for this experiment can be viewed at the anonymous link below:
> >
> > [Sampled prompts transfer results](https://drive.google.com/file/d/1JQNhSSLdzC96cChSrjoMWUV0qI5rr58V/view?usp=sharing)
> >
> > Transfer succeeds in nearly all cases, confirming that transferable solutions exist beyond discrete prompts.
> >
> > ## Addressing Limitations: Breadth of Transfer Functions
> >
> > One limitation of the original study is that we primarily explored a linear Transfer Function that minimized a Least Squares objective. We now address this limitation in this section, and show that findings in Section 4 are robust to the design of the Transfer Function. We illustrate this by making two modifications to the original Transfer Function, which minimized a least squares loss (Equation 2).
> >
> > * **Modification 1 - Adding sparse regularization**
> >
> > To begin, we have reproduced Figure 4 of Section 4, using a sparse regularization term that penalizes the L1 norm of the transformation matrix T added to the L2 loss.. Specifically, the objective is:
> >
> > $\arg \min_{T} \; \mathbb{E} \left\| \vec{x}(w) - T \vec{y}(w) \right\|^2_2 + \lambda \left\| T \right\|_1$
> >
> > Results for this ablation can be viewed here:
> >
> > [Sparse regularization](https://drive.google.com/file/d/1R7a9gm5jYAtTuQ6x6K90kiRng0P6UiJ4/view?usp=sharing)
> >
> > For all tested models and datasets, results agree with original findings.
> >
> > * **Modification 2 - Changing to L1 loss**
> >
> > For completeness, we conduct a second ablation replacing the original least squares loss with an L1 loss instead---this time with no sparse regularization applied. The objective in this ablation is:
> >
> > $\arg \min_{T} \; \mathbb{E} \left\| \vec{x}(w) - T \vec{y}(w) \right\|_1$
> >
> > Results for this ablation can be viewed here:
> >
> > [L1 loss](https://drive.google.com/file/d/1vKf97_79Wsqi5OYbiuOZwyzxIYj85Dm9/view?usp=sharing)
> >
> > Findings in both ablations agree with the original findings, suggesting that **conclusions drawn in our study are not impacted by the Transfer Function,** and are deeper properties of the underlying models.
> >
> > * **Nonlinear transfer functions**
> >
> > We also highlight *Appendix H, Figure 9* using a two-layer MLP Transfer function. Results in this ablation are consistent with the two modifications provided above, and reinforce the existing message of Section 4.

---

> ### Author Response · Authors · 2024-11-23
> **Response To Reviewer 2W9n 2/4**
>
> ## Clarifying Our Motivation and Goal
>
> Parts of the review focus on the transferability experiments, but these are one component of a broader study that aims to understand how solutions found via prompt tuning methods differ from traditional discrete prompts. Based on results in the manuscript, and new experiments in this rebuttal, they differ in two key ways:
>
> **(Property 1)**: Prompt tuning solutions are non-transferable, despite base input embeddings exhibiting high semantic similarity across domains, and despite many linearly transferable embeddings existing.
>
> **(Property 2)**: Prompt tuning solutions target the final layers in models.
>
> In this section, we explore **(Property 1)** by measuring the semantic similarity of the input embeddings of tested models via the mutual nearest neighbors of shared tokens.
>
> * **High Semantic Similarity According to Mutual Nearest Neighbors**
>
> To approach this experiment, we adapt the Mutual Nearest Neighbors metric (Mnn) from “The Platonic Representation Hypothesis” by Hut et al. 2024 [1]. To ensure that our measured value for this metric is directly comparable to results reported in the original paper from Hut et al. 2024, we employ their hyperparameters.
>
> | Task A     | Task B         |      Mnn |
> |:-----------|:---------------|---------:|
> | generation | detection      | 0.21537  |
> | generation | classification | 0.164991 |
> | detection  | classification | 0.157687 |
>
> The baseline similarity between Dinov2 and Llama3 is `0.16`, and values for the models we tested are as high or higher than this baseline, **suggesting base embeddings for all tested models are semantically similar.**
>
> * **Considering An Alternative Metric**
>
> For completeness, we note a potential limitation of the Mnn metric---euclidean distance is perhaps more useful for general representations than for input embeddings, which are often spherically distributed. Cosine similarity may be more suitable to compare input embedding spaces than euclidean distance, so we re-compute the Mnn metric using cosine similarity instead of euclidean distance in the following table.
>
> | Task A     | Task B         |      Mnn |
> |:-----------|:---------------|---------:|
> | generation | detection      | 0.350069 |
> | generation | classification | 0.35019  |
> | detection  | classification | 0.320711 |
>
> Results show that for the k = 10 nearest neighbors according to the cosine similarity, on average roughly `35%` of the neighboring tokens are the same across all three classes of models.
>
> * **Interpreting The Findings**
>
> Given that all tested models have embedding spaces with similar structure, it is perhaps surprising that models with similarly structured embedding spaces have incompatible prompt tuning solutions.
>
> [1] The Platonic Representation Hypothesis, Huh, Minyoung, et al., ArXiv 2024.

---

> ### Author Response · Authors · 2024-11-23
> **Response To Reviewer 2W9n 4/4**
>
> ## Miscellaneous Points
>
> **”What if you were to reverse the setup, learn prompts for discriminative tasks and transfer to generative tasks would the results hold?”**
>
> We explore all six transfer directions between the three tested model families. Performance for all six transfer directions can be viewed in Section 4, Figure 4 of the manuscript, where lines labeled “Trained For Task A” in the row for Task B indicate the performance of transferring prompts from Task A to Task B.
>
> **”Did you finetune any layers of the models? To the best of my knowledge, they seemed to have been left frozen.”**
>
> No models were fine-tuned, only the prompt embeddings were tuned. This is an important constraint to ensure that our findings apply to the original models as they would be used by researchers in the field.
>
> **”The vocabulary in some sections feels unnecessarily complex and could be simplified for better clarity.”**
>
> Thank you for your feedback, we are revising and simplifying the phrasing of the manuscript to improve readability and clarity.

---

> ### Author Response · Authors · 2024-11-26
> **Reminder to Reviewer 2W9n**
>
> Dear Reviewer, thank you for your initial feedback on the manuscript. We believe our rebuttal provides a new perspective on the questions raised in your review, and addresses potential limitations comprehensively.
>
> The rebuttal provides new experiments and clarifications, including:
>
> * **The Motivation Of This Paper And Its Findings**
> * **Why The Transfer Setup Is Valuable: It Reveals a Surprising Property**
> * **Addressing Limitations: Breadth of Transfer Functions**
>
> We are grateful for your time, and if any questions remain, we hope to continue the discussion.

---

> > ### Comment · Reviewer_2W9n · 2024-11-26
> >
> > Hi Authors,
> >
> > Thank you for your detailed response to my questions in my review. I now realize that I had slightly misunderstood the motivation of this paper. I see now that it is intended to be more of an analysis-driven work, focusing on how different prompt-learning solutions can be effectively adapted across various tasks and models.
> >
> > I believe these insights are both novel and valuable for the community. However, I feel that the focus on transferability is largely a consequence of the closed-source nature of many Vision-Language Models (VLMs). In an ideal scenario, where these models and their pretraining data are fully open-source, the need to develop transferable soft prompts across models would be not needed.
> >
> > The authors also perform different ablations and experiements in this rebuttal that answers my questions, and in that case I have decided to bump my score upto 6

---

### Official Review · Reviewer_2bvc · 2024-11-01

**Soundness:** 2
**Presentation:** 3
**Contribution:** 1
**Rating:** 5
**Confidence:** 1

**Summary:**

This work studies the following question: How are prompt embedding for visual concepts found by prompt tuning methods different from typical discrete prompts? It evaluates through transfer learning tasks such as zero-shot detection.

**Strengths:**

The paper is well-written (while I have a hard time to understand the motivation, see below) but the overall presentation is good. Authors did a large-scale analysis on the defined problem as well.

**Weaknesses:**

I actually have a hard time to understand the motivation of this work, and as a result, my judgement may be incorrect.

Specifically, it's not clear to me what's the motivation of finetuning prompts like <black_dog> or <orange_cat>? I have seen work doing similar things for personalized generation like DreamBooth but what's actually the motivation for prompts like <black_dog> or <orange_cat>?

Following previous point, I don't get the motivation to understand the difference between <black_dog> and "black dog" in the embedding space. For what purposes should we care about this? I think the analysis makes sense but it's not clear to me why should we care about this problem.

**Questions:**

See above

---

### Official Review · Reviewer_52QW · 2024-11-01

**Soundness:** 2
**Presentation:** 2
**Contribution:** 2
**Rating:** 6
**Confidence:** 3

**Summary:**

The paper investigates prompt embeddings in large multimodal models, such as Stable Diffusion, to understand how these embeddings differ from traditional discrete prompts for generating and classifying new visual concepts. Through a large-scale analysis across text-to-image generation, object detection, and zero-shot classification, the authors discover that prompts optimized for new concepts function similarly to adversarial attacks on the text encoder. Testing with 4,800 embeddings, they find that these adversarial perturbations specifically target the final layers of text encoders, influencing models to respond to specific subjects, but these effects are model-specific and dependent on initialization.

**Strengths:**

- The paper presents an interesting perspective for integrating specific concepts into a sequence of multimodal models.
- The paper proposed straightforward methods for transferring the soft prompts in a source domain into various target tasks, with thorough analysis of its effect.

**Weaknesses:**

- The paper presumes the linearity between the text embedding space between domains. While models sharing the similar text encoders might be suitable to presume the linear relationship, the models using text encoders with totally different text embedding space might rather collapse when representing the soft prompt of the target domain with linearity.
- Generalizability of transform function: the paper used 40 visual concepts for transferring experiments, which seem to be limited. Scaling up the visual concepts would be required to see if the transform function can generalize to any of visual concepts.
- the motivation behind transferring the soft prompt over various tasks: the authors suggested that transferring the soft prompt into other tasks eliminates the need for retraining prompts for each task. Performance comparison between transferring prompt vs prompt-tuning each task would be required to see if the performance gap is negligible while lowering the training overhead.
- The paper needs re-ordering, where the main goal of why transferring soft prompts is needed stated in the last with a few sentences, which might give not seamlessly connected
- The paper needs reorganization, as the empirical questions and observations take a large portion of the introduction and abstract in the front, while the main goal—explaining why transferring soft prompts is necessary—is briefly mentioned at the end with a few sentences, which might make the flow disjointed. Also, the details (e.g., derivation of (2)) are often absent, which would rather be better if kindly provided in supplementary materials.

**Questions:**

See above weaknesses.

---

> ### Author Response · Authors · 2024-11-22
> **Response To Reviewer 52QW 1/4**
>
> Thank you for your feedback on the manuscript, there are several points made in the review: (1) Are the input embeddings of models in different domains similarly structured? (2) Do the input embeddings of models in different domains exhibit a linear relationship? (3) Do the findings in our study generalize to diverse concepts? (4) Why is transfer an important tool for understanding prompt tuning solutions?
>
> We conduct new experiments and ablations that address these points:
>
> ## Response Summary
>
> * **Addressing point (1)**: We conduct an experiment using the Mutual Nearest Neighbors metric (Mnn) proposed in “The Platonic Representation Hypothesis” by Hut et al. 2024. Using the same hyperparameters as their work to ensure that values are directly comparable, we find a semantic similarity between `0.157 - 0.215`, which *exceeds the semantic similarity of Dinov2 and Llama3 from Hut et al. 2024*, suggesting a high similarity.
>
> * **Addressing point (2)**: We have added a control experiment to the paper, showing a regime where a linear transfer function successfully maps performant solutions between two spaces. Linearly transferable vector embeddings exist for all tested models, suggesting that linearity is not a limiting factor in our study.
>
> * **Addressing point (3)**: We have added experiments on the EuroSAT dataset, a remote sensing task with 10 diverse concepts from satellite imagery of different geographic features. Results on EuroSAT are consistent with our main findings, and show that conclusions from our study generalize to this more challenging domain.
>
> * **Addressing point (4)**: We think our goal may be misunderstood. Our goal is to understand how solutions found via prompt tuning methods differ from traditional discrete prompts, and the non-transferability problem we discover is not the only result of our pursuit of this question. We also identify that prompt tuning solutions have a second property in which they differ from traditional discrete prompts: they target the final layers in models.
>
> We believe the new experiments added in this rebuttal show that even models with similarly structured embedding spaces have incompatible prompt tuning solutions. This highlights the importance of a paper dedicated solely to understanding **how prompt tuning solutions differ from discrete prompts.**
>
> Additional discussion for these points is provided below.

---

> ### Author Response · Authors · 2024-11-22
> **Response To Reviewer 52QW 2/4**
>
> ## Base Embeddings are Semantically Similar
>
> The first point in the review pertains to whether the input embedding spaces are semantically similar. We agree with the reviewer that similarity is an important requirement, as high similarity makes our findings more surprising---that models with similarly structured embedding spaces have incompatible prompt tuning solutions.
>
> * **High Semantic Similarity According to Mutual Nearest Neighbors**
>
> To approach this experiment, we adapt the Mutual Nearest Neighbors metric (Mnn) from “The Platonic Representation Hypothesis” by Hut et al. 2024 [1]. To ensure that our measured value for this metric is directly comparable to results reported in the original paper from Hut et al. 2024, we employ their hyperparameters.
>
> | Task A     | Task B         |      Mnn |
> |:-----------|:---------------|---------:|
> | generation | detection      | 0.21537  |
> | generation | classification | 0.164991 |
> | detection  | classification | 0.157687 |
>
> The baseline similarity between Dinov2 and Llama3 is `0.16`, and values for the models we tested are as high or higher than this baseline, **suggesting base embeddings for all tested models are semantically similar.**
>
> * **Considering An Alternative Metric**
>
> For completeness, we note a potential limitation of the Mnn metric---euclidean distance is perhaps more useful for general representations than for input embeddings, which are often spherically distributed. Cosine similarity may be more suitable to compare input embedding spaces than euclidean distance, so we re-compute the Mnn metric using cosine similarity instead of euclidean distance in the following table.
>
> | Task A     | Task B         |      Mnn |
> |:-----------|:---------------|---------:|
> | generation | detection      | 0.350069 |
> | generation | classification | 0.35019  |
> | detection  | classification | 0.320711 |
>
> Results show that for the k = 10 nearest neighbors according to the cosine similarity, on average roughly `35%` of the neighboring tokens are the same across all three classes of models.
>
> * **Interpreting The Findings**
>
> Given that all tested models have embedding spaces with similar structure, it is perhaps more surprising that models with similarly structured embedding spaces have incompatible prompt tuning solutions.
>
> [1] The Platonic Representation Hypothesis, Huh, Minyoung, et al., ArXiv 2024.
>
> ## Control Experiment for the Transfer Function
>
> A second point in the review pertains to the effectiveness of the Transfer Function. To address this question, we conduct an experiment showing cases where transfer succeeds, and transferred embeddings attain high performance, proving these solutions exist, but that prompt tuning does not recover these solutions.
>
> We show two successful linearly transferable scenarios:
>
> * **Transferring discrete prompts**
>
> In this experiment, we take the embeddings for tokens of the class name for the target consent (i.e. “sombrero” for the sombrero class), and we transfer embeddings between models following the methodology in Section 4. Results for this experiment can be viewed at the anonymous link below:
>
> [Discrete prompts transfer results](https://drive.google.com/file/d/1yP7_DTPpJaos195l5bSQdOZ6plZDGRsl/view?usp=sharing)
>
> Transfer succeeds in all cases, suggesting the Transfer Function is an effective map.
>
> * **Transferring sampled prompts**
>
> Following up the previous experiment, we conduct a second experiment where we sample embeddings in the neighborhood of embeddings for tokens of the class name for the target consent. In particular, we employ a normal distribution centered at the embedding for tokens of class names, with a standard deviation proportional to the distance between tokens and their closest neighbor (so samples stay in their original neighborhood).
>
> Results for this experiment can be viewed at the anonymous link below:
>
> [Sampled prompts transfer results](https://drive.google.com/file/d/1JQNhSSLdzC96cChSrjoMWUV0qI5rr58V/view?usp=sharing)
>
> Transfer succeeds in nearly all cases, confirming that transferable solutions exist beyond discrete prompts.

---

> ### Author Response · Authors · 2024-11-22
> **Response To Reviewer 52QW 3/4**
>
> ## Diversity and Representativeness of Datasets
>
> We selected four standard computer vision tasks employed by researchers as they develop applications with Large Multimodal Models. In particular, ImageNet is frequently used with CLIP, the DreamBooth dataset is frequently used with Stable Diffusion, and COCO + Pascal are frequently used with Owl-v2.
>
> The purpose in selecting these datasets is to ensure that tasks are representative of how the selected models are currently used by researchers in the field. One challenge we face when selecting datasets is coverage---all selected datasets have target subjects to generate, aligned class labels, and instance bounding boxes.
>
> This constraint allows us to transfer embeddings between all families of models, which is an important feature of the study to ensure that findings are generalizable, and not specific to one model family.
>
> * **New Results on EuroSAT**
>
> Based on your request for more diverse concepts, we have added the EuroSAT dataset to our study (a remote sensing task), and results can be viewed at the following anonymous link:
>
> [EuroSAT transfer results](https://drive.google.com/file/d/10kqgIDK0aK2rKHJXSst-ixsQjAfFBJNq/view?usp=sharing)
>
> Note that EuroSAT was not originally considered for inclusion in our study because it lacks instances that can be used to study transfer with object detection models. We thus only consider transfer between generation and classification models on EuroSAT, and show that findings on EuroSAT match our original findings.
>
> The inclusion of EuroSAT improves the diversity of tasks in our study (from 40 $\to$ 50 concepts), and the consistency of findings reinforces that conclusions drawn are deeper properties of models.

---

> ### Author Response · Authors · 2024-11-22
> **Response To Reviewer 52QW 4/4**
>
> ## Miscellaneous Points
>
> **”the main goal—explaining why transferring soft prompts is necessary—is briefly mentioned at the end with a few sentences, which might make the flow disjointed”**
>
> We think the main goal of this study is misunderstood. Our main goal is to understand how solutions found via prompt tuning methods differ from traditional discrete prompts, and transfer is a probative tool for this purpose.
>
> We are revising the referenced sentence from our paper’s introduction to clarify our goal.
>
> We believe the new experiments added in this rebuttal show that even models with similarly structured embedding spaces have incompatible prompt tuning solutions. This highlights the importance of a paper dedicated solely to understanding **how prompt tuning solutions differ from discrete prompts.**
>
> **”Performance comparison between transferring prompt vs prompt-tuning each task would be required”**
>
> This comparison is presented in Section 4, Figure 4 of the manuscript, where lines labeled “Trained For Task A” in the row for Task B indicate the performance of transferring prompts from Task A to Task B.
>
> **”Also, the details (e.g., derivation of (2)) are often absent, which would rather be better if kindly provided in supplementary materials.”**
>
> The referenced equation from the paper is the standard Least Squares formulation [2], which involves the minimization of a quadratic cost function by taking the derivative with respect to the linear transformation matrix T, setting the derivative equal to zero, and solving for the optimal transformation matrix T.
>
> We are adding the derivation of the least squares solution to the Appendix based on your feedback.
>
> [2] Introduction to Applied Linear Algebra – Vectors, Matrices, and Least Squares
> Stephen Boyd and Lieven Vandenberghe, Cambridge University Press.

---

> ### Author Response · Authors · 2024-11-26
> **Reminder to Reviewer 52QW**
>
> Dear Reviewer, thank you for your initial feedback on the manuscript. We value your impression, and have taken care to address the questions raised in your review with a comprehensive rebuttal.
>
> The rebuttal provides new experiments and clarifications on:
>
> * **The Motivation Of This Paper And Its Findings**
> * **Are The Base Input Embeddings Similarly Structured Across Models**
> * **Examples Where Embeddings Are Linearly Transferable**
> * **Diversity and Representativeness of Datasets**
>
> We hope the additional experiments and clarifications help provide a deeper understanding of our contributions in this work, and their importance. If any questions remain or if further clarifications would be helpful, we are grateful for the opportunity to continue this discussion.
>
> Best, The Authors

---

> ### Comment · Reviewer_52QW · 2024-11-26
>
> Thanks for the detailed responses. All my concerns were resolved, so I raised my score.

---

### Official Review · Reviewer_nh4x · 2024-11-02

**Soundness:** 3
**Presentation:** 3
**Contribution:** 3
**Rating:** 6
**Confidence:** 3

**Summary:**

This paper examines how fine-tuned prompt embeddings for visual concepts affect text-to-image generation, object detection, and classification. It reveals that these embeddings act as model-specific adversarial perturbations, altering behavior without needing extensive retraining.

**Strengths:**

- This paper provides a novel perspective on prompt tuning across multiple models and tasks.
- The paper is logically structured, clearly presenting methodologies and findings.

**Weaknesses:**

- The application of the transfer function and its relationship with Section 4 should be clearly articulated, especially regarding its role in the findings and conclusions of the study.
- The potential impact of the differences in the datasets used for the experiments on the results and conclusions should be discussed in detail. Currently, the datasets appear to exhibit considerable similarity. The diversity and representativeness of these datasets remain limited. It raises the question of whether the findings from a model trained on large-scale data are genuinely necessary for the conclusions drawn in this study. The application of the findings in domains with greater diversity could yield more valuable insights. A comparative analysis that includes varied domains, such as remote sensing, would enhance our understanding of the generalizability of the findings.

**Questions:**

- The analysis of the transfer function employs a straightforward linear transformation.
1. Given that X and Y are finite observations, the effectiveness of the linear transformation is influenced by the sample size (n), quality, and representativeness of the samples. Under these circumstances, it is critical to evaluate whether the derived transfer function T is representative and adequately supports the subsequent conclusions.
2. Considering the potential noise in the observations, it would be worthwhile to explore the use of penalized least squares. Additionally, for linear transformations between two spaces, incorporating sparse regularization could significantly impact the subsequent analysis and conclusions.

---

> ### Author Response · Authors · 2024-11-22
> **Response To Reviewer nh4x 1/3**
>
> Thanks for your feedback on the manuscript. Several points are discussed in the review, and addressed in this rebuttal, including: (1) Impact of the transfer function on the findings in Section 4, (2) Diversity and representativeness of datasets included in the study, (3) Effectiveness of the transfer function.
>
> We conduct new experiments and ablations that address these points:
>
> ## Response Summary
>
> * **Addressing point (1)**: We have conducted a series of ablations on the transfer function, including sparsity regularization based on an L1 penalty on the linear transformation matrix T, and a different loss function, replacing the original L2 loss with the L1 loss. Findings are not impacted by these changes.
>
> * **Addressing point (2)**: We have added EuroSAT based on your suggestion, a remote sensing dataset that includes 10 visual concepts representing satellite imagery of different geographic features. Results on EuroSAT support findings discovered on the original four datasets.
>
> * **Addressing point (3)**: We have added a control experiment to the paper, showing a regime where the transfer function successfully maps performant solutions between two spaces. Transferable vector embeddings exist for all tested models, suggesting the Transfer Function is not the limiting factor.
>
> Additional discussion for these points is provided below.

---

> ### Author Response · Authors · 2024-11-22
> **Response To Reviewer nh4x 2/3**
>
> ## Impact of Transfer Function on Findings
>
> Findings in Section 4 are robust to the design of the Transfer Function. We illustrate this by making two modifications to the original Transfer Function, which minimized a least squares loss (Equation 2).
>
> * **Modification 1 - Adding sparse regularization**
>
> Based on your suggestion, we have reproduced Figure 4 of Section 4, using a sparse regularization term that penalizes the L1 norm of the transformation matrix T added to the L2 loss.. Specifically, the objective is:
>
> $\arg \min_{T} \; \mathbb{E} \left\| \vec{x}(w) - T \vec{y}(w) \right\|^2_2 + \lambda \left\| T \right\|_1$
>
> Results for this ablation can be viewed here:
>
> [Sparse regularization](https://drive.google.com/file/d/1R7a9gm5jYAtTuQ6x6K90kiRng0P6UiJ4/view?usp=sharing)
>
> For all tested models and datasets, results agree with original findings.
>
> * **Modification 2 - Changing to L1 loss**
>
> For completeness, we conduct a second ablation replacing the original least squares loss with an L1 loss instead---this time with no sparse regularization applied. The objective in this ablation is:
>
> $\arg \min_{T} \; \mathbb{E} \left\| \vec{x}(w) - T \vec{y}(w) \right\|_1$
>
> Results for this ablation can be viewed here:
>
> [L1 loss](https://drive.google.com/file/d/1vKf97_79Wsqi5OYbiuOZwyzxIYj85Dm9/view?usp=sharing)
>
> Findings in both ablations agree with the original findings, suggesting that **conclusions drawn in our study are not impacted by the Transfer Function,** and are deeper properties of the underlying models.
>
> * **Nonlinear transfer functions**
>
> We also highlight *Appendix H, Figure 9* using a two-layer MLP Transfer function. Results in this ablation are consistent with the two modifications provided above, and reinforce the existing message of Section 4:
>
> *(Point A)* Prompt tuning finds fractured solutions.
>
> *(Point B)* One property of these solutions is they are non-transferable.
>
> *(Point C)* Another property of fractured solutions is they target specific layers in the models.
>
> We are happy to continue the discussion if other questions remain.

---

> ### Author Response · Authors · 2024-11-22
> **Response To Reviewer nh4x 3/3**
>
> ## Diversity and Representativeness of Datasets
>
> We selected four standard computer vision tasks employed by researchers as they develop applications with Large Multimodal Models. In particular, ImageNet is frequently used with CLIP, the DreamBooth dataset is frequently used with Stable Diffusion, and COCO + Pascal are frequently used with Owl-v2.
>
> The purpose in selecting these datasets is to ensure that tasks are representative of how the selected models are currently used by researchers in the field. One challenge we face when selecting datasets is coverage---all selected datasets have target subjects to generate, aligned class labels, and instance bounding boxes.
>
> This constraint allows us to transfer embeddings between all families of models, which is an important feature of the study to ensure that findings are generalizable, and not specific to one model family.
>
> * **New Results on EuroSAT**
>
> Based on your feedback, we have added the EuroSAT dataset (a remote sensing task), and results can be viewed at the following anonymous link:
>
> [EuroSAT transfer results](https://drive.google.com/file/d/10kqgIDK0aK2rKHJXSst-ixsQjAfFBJNq/view?usp=sharing)
>
> Note that EuroSAT was not originally considered for inclusion in our study because it lacks instances that can be used to study transfer with object detection models. We thus only consider transfer between generation and classification models on EuroSAT, and show that findings on EuroSAT match the original findings.
>
> The inclusion of EuroSAT improves the diversity of tasks in our study, and the consistency of findings reinforces that conclusions drawn in our paper are properties of the underlying models.
>
> ## Effectiveness of Transfer Function
>
> The final question in the review pertains to the effectiveness of the Transfer Function. To address this question, we conduct an experiment showing cases where transfer succeeds, and transferred embeddings attain high performance, proving these solutions exist, but that prompt tuning does not recover these solutions.
>
> We show two successful transfer scenarios:
>
> * **Transferring discrete prompts**
>
> In this experiment, we take the embeddings for tokens of the class name for the target consent (i.e. “sombrero” for the sombrero class), and we transfer embeddings between models following the methodology in Section 4. Results for this experiment can be viewed at the anonymous link below:
>
> [Discrete prompts transfer results](https://drive.google.com/file/d/1yP7_DTPpJaos195l5bSQdOZ6plZDGRsl/view?usp=sharing)
>
> Transfer succeeds in all cases, suggesting the Transfer Function is an effective map.
>
> * **Transferring sampled prompts**
>
> Following up the previous experiment, we conduct a second experiment where we sample embeddings in the neighborhood of embeddings for tokens of the class name for the target concept. In particular, we employ a normal distribution centered at the embedding for tokens of class names, with a standard deviation proportional to the distance between tokens and their closest neighbor (so samples stay in their original neighborhood).
>
> Results for this experiment can be viewed at the anonymous link below:
>
> [Sampled prompts transfer results](https://drive.google.com/file/d/1JQNhSSLdzC96cChSrjoMWUV0qI5rr58V/view?usp=sharing)
>
> Transfer succeeds in nearly all cases, confirming that transferable solutions exist beyond discrete prompts.

---

> > ### Comment · Reviewer_nh4x · 2024-11-25
> >
> > I would like to thank the authors for responding. All the responses have addressed my primary concerns. I keep the score as it is.

---

### Author Response · Authors · 2024-11-24
**Reminder To Reviewers, and Summary of Rebuttal**

Dear Reviewers,

We hope this message finds you well. We are writing to remind that the rebuttal phase for our submission, *Understanding Visual Concepts Across Models*, will close in two days. We deeply appreciate your initial feedback, which has aided in refining our work. In response, we provide a thorough rebuttal with several new experiments aimed at addressing your concerns, including:

* **Transfer Function Ablations**
* **Control Experiments Where Transfer Works**
* **Confirming High Structural Similarity of Base Embeddings**
* **Additional Datasets To Improve Diversity**

We believe our rebuttal provides a new perspective on the questions raised and addresses potential limitations comprehensively. If you could take a moment to review the rebuttal and share any feedback or updated impressions, we would be grateful. Please do not hesitate to let us know if there are any remaining questions or concerns that need further clarification.

Best, The Authors

---

### Meta-Review · Area_Chair_U3Rz · 2024-12-20

**Metareview:**

The paper examines prompt embeddings in large multimodal models like Stable Diffusion, revealing that optimized prompts (via prompt-tuning) resemble adversarial attacks on text encoders. Analyzing 4,800 embeddings across tasks, the study shows these perturbations target the final text encoder layers, directing models toward specific subjects. They also find that perturbations reprogramming multimodal models are initialization-specific, and model-specific.

Except for reviewer 2bvc whose confidence score is 1, all other reviewers unanimously agree to accept this paper.

This paper provides a novel observation about prompt tuning approaches that are used to capture new concepts. This would be a interesting finding for the general community. So I recommend accept.

**Additional Comments On Reviewer Discussion:**

Most concerns have been properly addressed after rebuttal.

---

### Decision · Program_Chairs · 2025-01-22

Accept (Poster)